# DeepAnalyze: Agentic Large Language Models for Autonomous Data Science

**Shaolei Zhang** [1]   **Ju Fan** [1]   **Meihao Fan** [1]   **Yizhe Liu** [1]   **Yuxin Zhang** [1]   **Xiaoyong Du** [1]

## Abstract

Autonomous data science has been a long-standing challenge, and is now becoming feasible with the emergence of powerful large language models (LLMs). Recent workflow-based data agents have shown promising results on specific data tasks but remain fundamentally limited in achieving full autonomy due to their reliance on predefined workflows. In this paper, we introduce DeepAnalyze, the first agentic LLM for autonomous data science, capable of automatically completing the end-to-end data science from raw data to research reports. To tackle high-complexity data science tasks, we propose a curriculum-based agentic training paradigm that emulates the learning trajectory of human data scientists, enabling LLMs to progressively acquire and integrate multiple capabilities in real-world environments. Accordingly, we contribute a data-grounded trajectory synthesis framework to construct high-quality data science training data. Through training in real-world environments, DeepAnalyze learns to perform a broad spectrum of data tasks, ranging from data question answering to open-ended data research. Experiments on 13 benchmarks demonstrate that, with only 8B parameters, DeepAnalyze outperforms workflow-based agents built on advanced proprietary LLMs. DeepAnalyze is fully open-sourced, paving the way toward autonomous data science[*].

## 1. Introduction

Autonomous data science ([Sun et al., 2025](); [Wang et al.,](); [2025](); [Rahman et al., 2025]()), a long-standing goal of the data

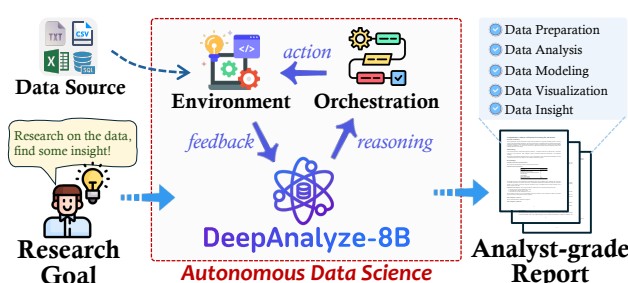

*Figure 1.* DeepAnalyze-8B is the first end-to-end agentic LLM that achieves autonomous data science, supporting the entire data science pipeline and open-ended data research.

science community, aims to automate the entire data science pipeline for extracting insights from data. However, the intrinsic complexity of data science has made this vision difficult to realize. Recent advances in large language models (LLMs) have demonstrated strong problem-solving capabilities ([DeepSeek-AI, 2025]()), reshaping paradigms in many areas. Despite this progress, most successes remain confined to unstructured inputs, and complex tasks involving structured data still lack a "one-size-fits-all" solution.

Handling complex structured data requires endowing LLMs with two higher-level capabilities: autonomous orchestration and adaptive optimization. Autonomous orchestration enables LLMs to understand user intent and coordinate sequences of interdependent actions to complete complex tasks ([Jeyakumar et al., 2024](); [Sapkota et al., 2025a]()). Adaptive optimization allows LLMs to interact with real-world data environments and iteratively refine their actions based on feedback ([Hong et al., 2025]()). As illustrated in Figure 1, an intelligent system equipped with these capabilities can support a wide range of tasks on structured data, from conventional question answering to open-ended data research.

Existing approaches to applying LLMs for autonomous data science can be broadly categorized into domain-specific LLMs and workflow-based agents. Early efforts primarily focus on developing domain-specific LLMs, such as code-oriented models ([Nascimento et al., 2024](); [Wen et al.,](); [2024]()) and structured data–oriented models ([Li et al., 2023b](); [Jiang et al., 2023](); [Xu et al., 2025]()), to address individual tasks including TableQA and code generation. However, these models lack capabilities for autonomous orchestration and adaptive optimization ([Yang et al., 2021](); [Li et al.,]()

---

[1]Renmin University of China. Correspondence to: Ju Fan (corresponding author) <fanj@ruc.edu.cn>.

*Proceedings of the $43^{rd}$ International Conference on Machine Learning*, Seoul, South Korea. PMLR 306, 2026. Copyright 2026 by the author(s).

[*] Code: https://github.com/ruc-datalab/DeepAnalyze
Model: https://huggingface.co/RUC-DataLab/DeepAnalyze-8B
Dataset: https://huggingface.co/datasets/RUC-DataLab/
DataScience-Instruct-500K

2023a), and therefore fall short of handling the entire data science pipeline. More recently, workflow-based data science agents (Hollmann et al., 2023; Guo et al., 2024; Hong et al., 2025) introduce predefined procedural workflows to prompt closed-source LLMs (e.g., GPT-4) to complete complex tasks. Although these approaches demonstrate strong performance, they rely heavily on manually designed heuristics, which limits their autonomy and adaptivity. In essence, both domain-specific LLMs and workflow-based agents remain constrained in their ability to perform complex tasks, as they are not trained in real-world environments, which has been shown to be crucial for enabling LLMs to develop autonomous problem-solving capabilities (Zheng et al., 2025; Jin et al., 2025).

In this paper, we aim to advance LLM-based data science methods from workflow-based agents to a trainable agentic model capable of autonomous exploration in real-world data environments. However, training such a LLM is non-trivial, as the high complexity of data science tasks results in severe reward sparsity and trajectory scarcity. Specifically, foundation LLMs often fail to complete complex data science tasks on structured data, resulting in extremely limited positive rewards, while the lack of long-horizon problem-solving trajectories provides no guidance for exploration, ultimately preventing LLMs from learning effectively in real-world data environments. To address these challenges, we introduce DeepAnalyze, the first agentic LLM designed for autonomous data science. To mitigate reward sparsity, DeepAnalyze introduces a curriculum-based agentic training paradigm inspired by the learning trajectory of human data scientists. This progressive easy-to-difficult schedule enables the model to gradually evolve from mastering individual abilities to developing comprehensive and adaptive problem-solving abilities in real-world environments. Meanwhile, we propose a data-grounded trajectory synthesis framework that automatically constructs high-quality reasoning and interaction trajectories, providing effective exploration guidance within a vast solution space. By combining curriculum-based training with trajectory synthesis, DeepAnalyze learns to autonomously orchestrate actions and adaptively optimize its strategies, thereby offering a general solution for data science tasks, from specific tasks to open-ended data research. Experimental results on 13 benchmarks show that, with only 8B parameters, DeepAnalyze-8B surpasses most advanced proprietary LLMs. More importantly, DeepAnalyze is the first agentic model capable of performing open-ended data research over structured data.

## 2. Related Work

**Autonomous Data Science.** Existing LLM-based data science approaches mainly fall into two categories: domain-specific LLMs and workflow-based agents. Early efforts focused on fine-tuning LLMs for individual data science tasks, including code generation for data analysis (Nascimento et al., 2024; Wen et al., 2024; Nejjar et al., 2024; Pan et al., 2025), tabular reasoning (Li et al., 2023b; Fang et al., 2024; Zhang et al., 2025c; Xu et al., 2025; Ouyang et al., 2025; Lei et al., 2025), and database interaction (Xue et al., 2024; Liu et al., 2024; Mohammadjafari et al., 2025). More recently, workflow-driven data agents have shown strong performance by progressively prompting LLMs to solve complex tasks (Hollmann et al., 2023; Guo et al., 2024; Yang et al., 2024; Hong et al., 2025; Ou et al., 2025; Maturo et al., 2025). These agents are typically built on Chain-of-Thought frameworks such as ReAct (Yao et al., 2023), Auto-Gen (Wu et al., 2024), and self-reflection (Pan et al., 2023), which decompose tasks into sequential subtasks. Despite differences in workflow design, most approaches depend on carefully engineered prompts to steer closed-source LLMs.

However, domain-specific LLMs are limited to isolated tasks, while workflow-based agents rely heavily on manually designed pipelines, preventing fully autonomous data science. In contrast, DeepAnalyze eliminates predefined workflows and prompting frameworks, instead internalizing data science capabilities through agentic training in real-world environments.

**Agentic Training for LLM.** Agentic training enhances LLMs via reinforcement learning, enabling multi-step reasoning and environment interaction (Plaat et al., 2025). It has shown practical success in coding (Sapkota et al., 2025b) and search (Zheng et al., 2025; Li et al., 2025; Jin et al., 2025). Existing methods typically regulate interaction formats through prompts and use final-answer accuracy as the RL reward, with lightweight cold-start techniques introduced to bootstrap format learning (DeepSeek-AI, 2025).

While effective for reasoning tasks, these approaches fall short for data science, which demands a broader skill set, including reasoning, structured data understanding, and code generation. Consequently, even cold-started LLMs struggle with complex data science problems, leading to reward sparsity and limited high-quality trajectories. To address this, we propose a curriculum-based agentic training framework that progressively develops data science abilities from single to multiple skills, supported by data-grounded trajectory synthesis to produce high-quality reasoning and interaction trajectories for training.

## 3. DeepAnalyze

In this paper, we introduce DeepAnalyze, an agentic large language model for autonomous data science. Inspired by the behavior of human data scientists, we first define a set of actions that enable DeepAnalyze to directly interact with the structured data. To endow the LLM with the capabil-

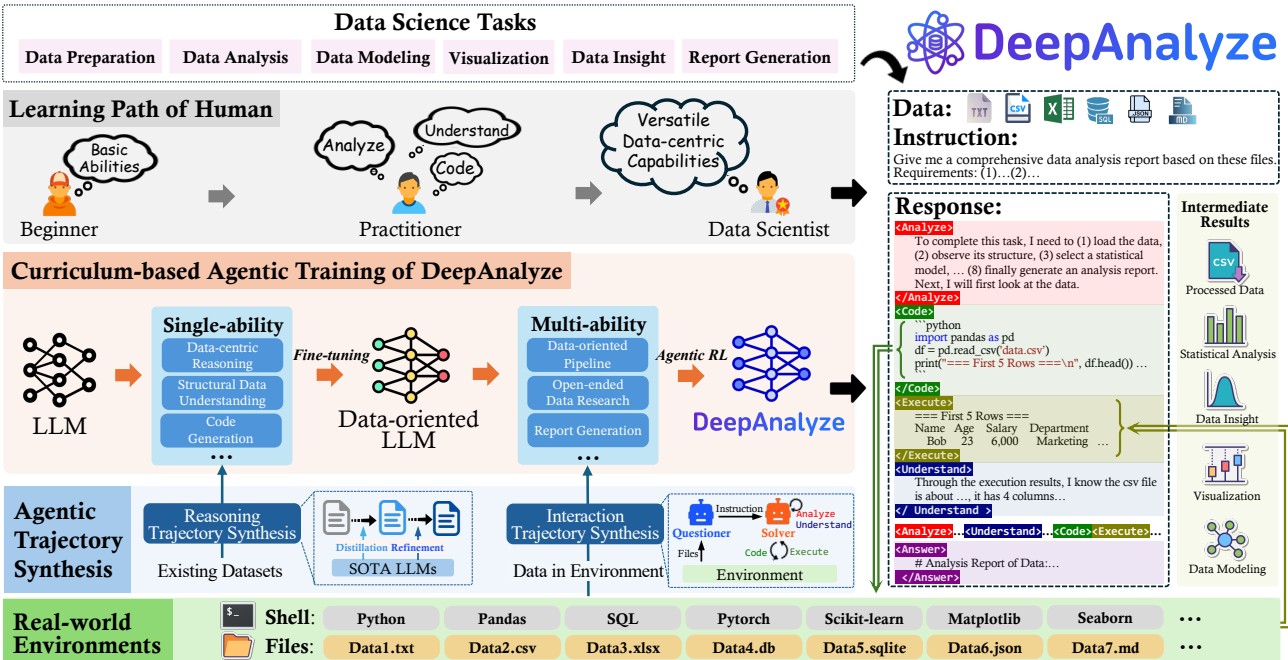

*Figure 2.* Architecture of DeepAnalyze.

ity for autonomous orchestration and adaptive optimization in real-world data environments, we propose a curriculum-based agentic training and data-grounded trajectory synthesis framework tailored for complex data science tasks. The architecture, curriculum-based agentic training, and data-grounded trajectory synthesis are introduced as follows.

### 3.1. Architecture

Unlike foundation LLMs that focus on understanding and generating natural language, LLMs for data science face the additional challenges of understanding and interacting with structured data, which is typically stored in external files. Therefore, DeepAnalyze extends natural language interaction with a data-oriented interaction pattern, thereby enabling LLMs to autonomously interact with real-world data environments.

**Input Format.** Previous structured data–specific LLMs (Li et al., 2023b; Fang et al., 2024; Zhang et al., 2025c; Xu et al., 2025; Lei et al., 2025) often converted tables stored in databases, CSV, or XLSX files into unstructured Markdown text and fed them into the LLM's context to enable structured data understanding. However, due to context length limitations, these methods can only handle small-scale data (e.g., very small tables). When human data scientists work with large-scale data, they do not passively read and memorize every record. Instead, they actively explore each data source as needed and then plan the following steps accordingly. To this end, DeepAnalyze integrates both modes: it passively accepts structured data expressed as text in the

input, while also actively inspecting external data sources according to user inputs, where the filenames of the external data sources are specified in the input, as shown in Figure 8.

**Interaction Pattern.** Given an instruction and the data sources in the environment, data scientists typically analyze, interact with the data in the environment, understand structured data, and iterate until the instruction is completed. To emulate this process, DeepAnalyze introduces five actions to automatically accomplish the data science task, including:

- $\langle$Analyze$\rangle \cdots \langle$/Analyze$\rangle$: Analyze textually, including planning, reasoning, reflection, self-verification...
- $\langle$Understand$\rangle \cdots \langle$/Understand$\rangle$: Understand the contents of data sources, such as databases, tables, and documents.
- $\langle$Code$\rangle \cdots \langle$/Code$\rangle$: Generate code to interact with the data in the environment, using Python suited for data science.
- $\langle$Execute$\rangle \cdots \langle$/Execute$\rangle$: Execute code and collect the feedback from the environment.
- $\langle$Answer$\rangle \cdots \langle$/Answer$\rangle$: Produce the final output.

In practice, we extend the vocabulary of the foundation LLM to support the generation of these special tokens. During inference, DeepAnalyze automatically switches between different actions by generating these special tokens, as shown in the right side of Figure 2. Specifically, once a $\langle$Code$\rangle \cdots \langle$/Code$\rangle$ is generated, DeepAnalyze executes the code in the environment and places the feedback in $\langle$Execute$\rangle \cdots \langle$/Execute$\rangle$, and then generates the next action. The detailed inference process of DeepAnalyze is shown in Algorithm 1. With this architecture, all actions (i.e., special tokens) are autonomously generated by the model without

---

**Algorithm 1** Inference of DeepAnalyze

---

1: **Input:** Instruction $Q$, Environment $Env$, DeepAnalyze model $\mathcal{M}$
2: **Output:** Response $A$ (with interaction process)
3: **Initialization:** $A = \emptyset$
4: **while** $\langle$Answer$\rangle \cdots \langle$/Answer$\rangle$ not in $A$ **do**
5:    $y \leftarrow \mathcal{M}(Q, A)$      // generate next action based on the instruction $Q$ and current response $A$
6:    $A \leftarrow A + y$
7:    **if** $\langle$Code$\rangle \cdots \langle$/Code$\rangle$ in $y$ **then**
8:       $code \leftarrow$ extract_code($y$)
9:       $feedback \leftarrow Env$.execute(code)   // interaction with the data in the environment
10:      $A \leftarrow A + \langle$Execute$\rangle + feedback + \langle$/Execute$\rangle$
11:    **end if**
12: **end while**
13: **Return** $A$

---

any human-defined workflows, which allows DeepAnalyze to fully autonomously orchestrate and optimize each action, laying the foundation for autonomous data science.

### 3.2. Curriculum-based Agentic Training

Under this architecture, DeepAnalyze must learn to interact with the environment to accomplish a wide range of data science tasks. Unlike isolated coding or retrieval tasks, data science problems require broader and more complex capabilities, including reasoning, structured data understanding, code generation, and the integrated skills needed to complete full data science pipelines and open-ended research. This complexity limits the proficiency of foundation LLMs in data science domains (Zhang et al., 2025b), leading to severe reward sparsity on complex tasks and rendering existing agentic training methods (such as RL-Zero or cold-start RL (DeepSeek-AI, 2025)) largely ineffective due to insufficient positive feedback. To address this challenge, we propose curriculum-based agentic training, which emulates the learning path of human data scientists by gradually transitioning from mastering single abilities to integrating multiple abilities. This training framework consists of two stages, where stage 1 employs single-ability fine-tuning to strengthen the foundation LLM's single ability, and stage 2 uses multiability agentic training to enable the LLM to apply multiple abilities in real-world environments to accomplish complex data science tasks.

**Single-ability Fine-tuning.** Since most foundation LLMs are not specifically trained for data science tasks, we first strengthen the core single abilities required for data science, including reasoning, structured data understanding, and code generation, corresponding to the actions $\langle$Analyze$\rangle$, $\langle$Understand$\rangle$, and $\langle$Code$\rangle$, respectively. Specifically, we fine-tune the foundation LLM on long CoT data (with reasoning traces) covering general reasoning tasks, structured data understanding, and code generation. This stage mirrors

the human learning process from a beginner to a data science practitioner in acquiring specialized skills, enhancing LLM's single ability in various aspects of data science.

**Multi-ability Agentic Training.** Building on the mastery of various single abilities, we employ agentic reinforcement learning to train DeepAnalyze to apply multiple abilities in real-world environments to complete complex data science tasks. To ensure the quality of reinforcement learning, we first perform a cold start by fine-tuning the LLM on synthesized interaction trajectories, enabling it to acquire basic capabilities in orchestrating and optimizing individual actions. Subsequently, we train DeepAnalyze in real-world environments using reinforcement learning with group relative policy optimization (GRPO) (Shao et al., 2024). For each question $q$ in training data $D$, GRPO samples a group of $G$ outputs $\{o_1, \cdots, o_G\}$ from the old policy $\pi_{\theta_{\text{old}}}$ and then optimizes the policy model $\pi_\theta$ by maximizing the following objective:

$$
\mathcal{J}_{\text{GRPO}}(\theta) = \mathbb{E}_{q \sim D, \{o_i\}_{i=1}^G \sim \pi_{\theta_{\text{old}}}(\cdot|q)} \left[ \frac{1}{G} \sum_{i=1}^G \left( \min\left( \frac{\pi_\theta(o_i|q)}{\pi_{\theta_{\text{old}}}(o_i|q)} A_i, \right. \right. \right.
$$
$$
\left. \left. \left. \text{clip}\left( \frac{\pi_\theta(o_i|q)}{\pi_{\theta_{\text{old}}}(o_i|q)}, 1 - \varepsilon, 1 + \varepsilon \right) A_i \right) - \beta \mathrm{D}_{KL}\left( \pi_\theta \parallel \pi_{\text{ref}} \right) \right) \right] \quad (1)
$$

where $A_i$ is the advantage calculated from the rewards $\{r_1, \cdots, r_G\}$ of outputs within each group, $\pi_{\text{ref}}$ is the reference model, $\varepsilon$ and $\beta$ are hyperparameters.

**Hybrid Reward Modeling.** The effectiveness of agentic reinforcement learning critically depends on both the training data and the reward function. We use the agentic interaction trajectories synthesized in Section 3.3 as training data, covering three broad categories of data science tasks: data question answering, specific data tasks (e.g., data preparation, analysis, visualization, modeling, and insight extraction), and open-ended research. Since many data science tasks are inherently open-ended, we adopt a hybrid reward-modeling approach that combines rule-based rewards with LLM-as-a-judge rewards. For all tasks, we first check whether the output format conforms to DeepAnalyze's architecture (i.e., whether it contains exactly five types of actions with the correct format). If the format is incorrect, we directly assign a reward of $R = -1$.

For data question answering and data-centric tasks, which have reference answers, the reward $R$ of each output $o$ is calculated using accuracy and interaction trajectory quality:

$$
R = \frac{1}{2}(\mathbb{1}_{acc}(o) + S_{interaction}(o)) \quad (2)
$$

where $\mathbb{1}_{acc}(o) \in \{0, 1\}$ indicates whether the result is correct, and $S_{interaction}(o) \in [0, 1]$ is a score to evaluate the quality of the interaction trajectory.

For open-ended research, the reward $R$ of each output $o$ is evaluated based on the quality of the final research report

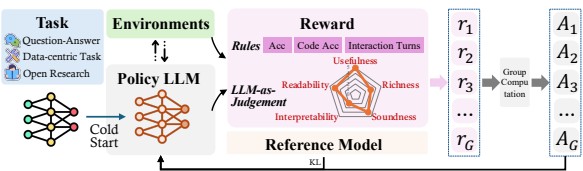

*Figure 3.* Schematic diagram of agentic RL.

*(a)* Reasoning Trajectory Synthesis *(b)* Interaction Trajectory Synthesis

*Figure 4.* Data-grounded trajectory synthesis on data science tasks.

and the research process. Denoting each interaction turn in output $o$ as $T_i \in o$, the reward $R$ is calculated as:

$$R = \frac{1}{3}\Big(S_{report}(o) + \min(\frac{|T|}{N^T}, 1) + \frac{1}{|T|}\sum_{T_i \in o} \mathbb{1}_{success}(T_i)\Big) \quad (3)$$

where $S_{report}(o)$ is the score that evaluates the generated report from five aspects: usefulness, richness, soundness, interpretability, and readability. $|T|$ measures the interaction turns with the environment, where $N^T = 10$ is a hyperparameter. $\mathbb{1}_{success}(T_i)$ indicates whether each interaction turn is successful. This reward encourages DeepAnalyze to engage in more successful interactions with the environment and to generate high-quality research reports.

Through curriculum-based agentic training, we progressively enhance DeepAnalyze's capabilities following an easy-to-hard schedule, enabling it to autonomously accomplish data science tasks in real-world environments.

### 3.3. Data-grounded Trajectory Synthesis

The proposed curriculum-based agentic training relies on high-quality reasoning and interaction trajectory data, while such data is unfortunately scarce for data science tasks. To overcome this challenge, we introduce a data-grounded trajectory synthesis framework that automatically constructs high-quality trajectory data tailored for data science tasks. The data-grounded trajectory synthesis framework consists of two parts: *Reasoning Trajectory Synthesis*, which constructs the reasoning trajectory for existing structured data instruction datasets, and *Interaction Trajectory Synthesis*, which constructs the entire data science trajectory based on structured data sources in the environment.

**Reasoning Trajectory Synthesis.** Existing instruction datasets for structured data, such as TableQA (Li et al., 2023b; Lei et al., 2025), structured knowledge grounding (Zhuang et al., 2024), and data science code generation, are useful to improve LLM's single capability. However, these datasets typically contain only instructions and responses, without the reasoning process. To address this limitation, we enhance existing datasets by synthesizing complex and refined reasoning trajectories, which are used for DeepAnalyze's single ability training.

As shown in Figure 4, reasoning trajectory synthesis builds upon the original instruction–response pairs through distillation and refinement. In the distillation stage, advanced LLMs are used as teacher models to extract reasoning tra-

jectories, whose correctness is verified by matching the generated responses with the ground-truth answers (DeepSeek-AI, 2025). To enhance structured data understanding, the distilled trajectories are reformulated into two complementary components: ⟨Analyze⟩ (reasoning process) and ⟨Understand⟩ (structured data comprehension). On this basis, we introduce keyword-guided refinement to further strengthen reasoning over structured data. Prior studies show that specific keywords (e.g., "but", "wait") play a critical role in reasoning (Zhang et al., 2025a; Shen et al., 2025). Motivated by this, we construct a key reasoning vocabulary and sample keywords to be inserted into the reasoning trajectories, encouraging deeper data-focused reasoning. Appendix G presents an example to illustrate how keyword insertion improves reasoning quality by directing attention to the underlying data. Overall, reasoning trajectory synthesis enables effective reuse of existing datasets, substantially enhancing DeepAnalyze's capabilities in reasoning, structured data understanding, and code generation.

**Interaction Trajectory Synthesis.** To enable DeepAnalyze to autonomously orchestrate and optimize multiple abilities in real-world environments, it is essential to construct multi-turn interaction trajectory data with the environment, yet such data is extremely scarce. In contrast, NL2SQL datasets such as Spider (Yu et al., 2018) and BIRD (Li et al., 2024) provide abundant structured data sources. To bridge this gap, we develop a multi-agent system to synthesize data science interaction trajectories from these data sources.

The multi-agent system involves three roles: questioner, solver, and inspector. The questioner observes the data sources in the environment and accordingly formulates a data science problem, conditioned on a sampled task type (e.g., data preparation, data analysis, data modeling, data insight, or open-ended research). Simultaneously, the questioner produces a checklist that serves as the evaluation criterion, including interaction-level constraints (e.g., number of turns, code library) and environment-level constraints (e.g., whether new files are generated, detailed file names). Given the data science problem and the data sources, the solver interacts with the environment using the introduced five actions to complete the task. Finally, the inspector validates the trajectory by checking the interaction process and environmental changes against the checklist, determining whether the trajectory should be accepted. Importantly, filtering trajectories based on both interaction details and environmental changes substantially improves the quality of

*Table 1.* Performance on DataSciBench. 'Success Rate' and 'Completion Rate' are pass rate and accuracy. 'VLM' and 'F1-F5' scores evaluate performance on various fine-grained data science sub-tasks, 'Score' denotes the overall performance.

| Models | Coarse-grained Metrics | | Fine-grained Metrics | | | | | | Score |
|---|---|---|---|---|---|---|---|---|---|
| | Success Rate | Completion Rate | VLM | F1: Data Preparation | F2: Plot Validity | F3: Data Exploration | F4: Data Visualization | F5: Data Modeling | |
| *Closed-Source API-Based Agent* | | | | | | | | | |
| **o1-mini** | 29.77 | 45.26 | 2.87 | 44.63 | 19.27 | 36.01 | 30.94 | 23.81 | 38.78 |
| **GPT-4o-mini** | 50.63 | 57.78 | 3.05 | 60.30 | 48.02 | 57.84 | 59.24 | 53.54 | 54.18 |
| **GPT-4o** | **66.31** | **68.44** | **3.91** | **75.93** | **56.14** | **69.33** | **71.35** | **57.67** | **64.51** |
| **GPT-4-Turbo** | 51.93 | 58.87 | 3.09 | 62.30 | 41.62 | 57.75 | 60.25 | 50.75 | 54.65 |
| **Claude-3-5-Sonnet** | 47.48 | 58.11 | 2.14 | 49.07 | 36.94 | 55.84 | 52.87 | 46.04 | 52.29 |
| **GLM-4-Flash** | 30.32 | 34.04 | 1.33 | 36.53 | 29.42 | 32.57 | 27.64 | 14.44 | 30.74 |
| *Open-Source LLM-based Agent* | | | | | | | | | |
| **Llama-3.1-8B-Instruct** | 24.73 | 33.89 | 1.29 | 38.24 | 18.25 | 21.98 | 22.89 | 25.85 | 29.69 |
| **Gemma-2-9B-it** | 7.07 | 11.00 | 1.06 | 26.16 | 16.90 | 23.81 | 18.11 | 17.15 | 12.66 |
| **GLM-4-9B-Chat** | 25.72 | 30.38 | 1.69 | 31.51 | 23.15 | 28.07 | 27.19 | 19.14 | 27.57 |
| **Qwen2.5-7B-Instruct** | 43.83 | 50.74 | 1.43 | 51.18 | 36.41 | 47.25 | 45.24 | 34.77 | 45.99 |
| **Qwen2-7B-Instruct** | 22.84 | 25.58 | 1.16 | 30.93 | 20.78 | 28.73 | 25.87 | 7.52 | 23.52 |
| **Yi-1.5-9B-Chat-16K** | 38.20 | 42.35 | 0.73 | 38.14 | 36.36 | 35.64 | 37.08 | 27.79 | 38.22 |
| **CodeLlama-13B-Instruct** | 10.49 | 14.64 | 0.04 | 11.67 | 11.34 | 9.43 | 14.43 | 5.15 | 12.64 |
| **CodeLlama-7B-Instruct** | 2.88 | 3.97 | 0.00 | 3.53 | 2.37 | 2.57 | 1.74 | 1.59 | 3.31 |
| **StarCoder2-15B** | 2.07 | 2.61 | 0.07 | 2.57 | 1.81 | 1.59 | 3.43 | 1.19 | 2.33 |
| **DeepSeek-Coder-6.7B-instruct** | 37.03 | 41.62 | 1.93 | 43.49 | 34.57 | 46.36 | 46.49 | 18.09 | 38.45 |
| **Qwen2.5-Coder-7B-Instruct** | 45.18 | 53.11 | 1.48 | 51.58 | 43.21 | 43.87 | 42.50 | **35.23** | 47.67 |
| *Agentic Model* | | | | | | | | | |
| **DeepAnalyze-8B** | **59.91** | **66.24** | 2.86 | **71.68** | **67.86** | **58.62** | **69.09** | 33.33 | **61.11** |

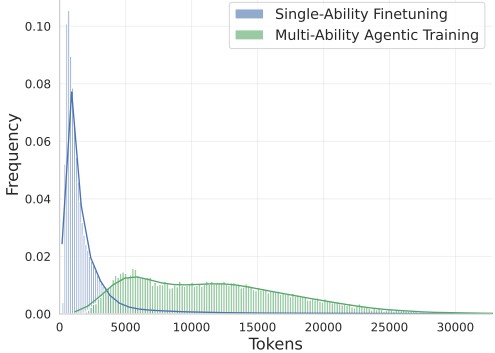

*Figure 5.* Length distribution of training data.

synthesized data. Through interaction trajectory synthesis, the high-quality multi-turn interaction data can be used for multi-ability agentic training (cold start and RL).

### 3.4. DataScience-Instruct-500K

We develop DeepAnalyze based on the constructed data in Sec.3.3. During the single-ability fine-tuning stage, we employ the reasoning trajectories built for data science, along with 100K general reasoning samples from AM-DeepSeek-R1-0528-Distilled[2]. In the multi-ability agentic training stage (including both cold start and RL phases), we use the interaction trajectories constructed for data science.

Figure 5 illustrates the length distribution of training data in both stages, with a sequence length of 8K in

the first stage and 32K in the second. In terms of scale, the single-ability fine-tuning stage consists of approximately 470K samples, the cold-start phase of multi-ability training includes 20K samples, and the RL phase comprises 15K samples, resulting in a total of around 500K samples. We release all training data, named `DataScience-Instruct-500K`[3], which can be used to train LLMs for data science tasks.

## 4. Experiments

### 4.1. Benchmarks

We conduct experiments on 13 data science benchmarks, including data science (**DataSciBench** (Zhang et al., 2025b), **DSBench** (Jing et al., 2025), **DABStep** (Egg et al., 2025), **DABStep-Research**, **KramaBench** (Lai et al., 2025)), code generation (**DS-1000** (Lai et al., 2023)) and structured data QA benchmarks (**WikiTQ** (Pasupat & Liang, 2015), **HybridQA** (Chen et al., 2020), **MultiHiertt** (Zhao et al., 2022), **OTT-QA** (Chen et al., 2021a), **FinQA** (Chen et al., 2021b), **TAT-QA** (Zhu et al., 2021), **HiTab** (Cheng et al., 2022)). Refer to Appendix D for detailed descriptions and metrics.

### 4.2. Experimental Setup

We build DeepAnalyze-8B based on DeepSeek-R1-0528-Qwen3-8B[4] as the foundation LLM. We use ms-swift (Zhao

---

[2] https://huggingface.co/datasets/a-m-team/AM-DeepSeek-R1-0528-Distilled

[3] https://huggingface.co/datasets/RUC-DataLab/DataScience-Instruct-500K

[4] https://huggingface.co/deepseek-ai/DeepSeek-R1-0528-Qwen3-8B

Table 2. Performance on DSBench (data modeling).

| Methods | LLM | Success (%) | Performance | Cost ($) |
|---|---|---|---|---|
| *Workflow-based Agent* | | | | |
| AutoGen | Llama3-8b | 5.41 | 1.55 | 0.00 |
| | Llama3-70b | 16.22 | 7.79 | 0.00 |
| | GPT-3.5 | 8.11 | 6.02 | 0.41 |
| | GPT-4 | 87.84 | **45.52** | 19.34 |
| | GPT-4o | 71.62 | 34.74 | 12.27 |
| | GPT-4o-mini | 22.97 | 11.24 | 0.10 |
| Code Interpreter | GPT-3.5 | 16.22 | 6.52 | 2.74 |
| | GPT-4 | 54.05 | 26.14 | 38.81 |
| | GPT-4o | 44.59 | 19.87 | 19.26 |
| | GPT-4o-mini | 39.19 | 16.90 | 2.70 |
| *Agentic Model* | | | | |
| **DeepAnalyze-8B** | | **90.63** | 39.41 | **0.00** |

Table 3. Performance on DABStep benchmark.

| Methods | LLM | Easy Level (72 Cases) | Hard Level (378 Cases) | Overall (450 Cases) |
|---|---|---|---|---|
| *Workflow-based Agent* | | | | |
| ReAct | Llama-4-Scout | 52.78 | 1.85 | 10.00 |
| | Qwen3-Coder | 54.17 | 3.44 | 11.56 |
| | GPT-4o-mini | 69.44 | 3.44 | 14.00 |
| | Deepseek-v3 | 66.67 | 5.56 | 15.34 |
| | GPT-4o | 66.67 | 6.08 | 15.77 |
| | Claude-3.5-Haiku | 77.78 | 5.03 | 16.67 |
| | Llama-4-Maverick | 75.00 | 8.73 | 19.33 |
| | GPT-4.1-mini | 77.78 | 8.99 | 20.00 |
| | Claude-3.5-Sonnet | 77.78 | 9.26 | 20.22 |
| | GPT-4.1 | 80.56 | 12.43 | 23.33 |
| Reasoning Prompt | o1 | 69.44 | 11.11 | 20.44 |
| | Gemini-2.5-Pro | 66.67 | 12.70 | 21.34 |
| | o3-mini | 72.22 | 13.76 | 23.11 |
| | o4-mini | 76.39 | 14.55 | 24.44 |
| DS-Agent | Gemini-2.0-Flash | 61.11 | 9.79 | 18.00 |
| Open Data Scientist | Deepseek-v3 | **84.72** | 16.40 | 27.33 |
| I2I-Agent | Claude-3.5-Sonnet | 80.56 | 28.04 | 36.44 |
| *Agentic Model* | | | | |
| **DeepAnalyze-8B** | | 70.83 | **32.80** | **38.88** |

et al., 2024) and SkyRL for training (refer to Appendix C for training parameters). The training data come from DataScience-Instruct-500K. During inference, we employ the vLLM engine (Kwon et al., 2023) for efficiency. All inference is performed within a sandbox environment to ensure the safety of code execution and data operations. The training and inference are conducted on NVIDIA A800 GPUs.

### 4.3. Main Results

**Capability on End-to-end Data Science Pipeline.** We evaluate DeepAnalyze on DataSciBench to assess its *end-to-end data science capabilities*, where each problem involves one or more sub-tasks such as data preparation, analysis, modeling, and visualization. We compare DeepAnalyze-8B with several workflow-based (ReAct) agents, covering 17 advanced LLMs. As shown in Table 1, coarse-grained metrics measure task success and sub-task completion, while fine-grained metrics assess performance across stages.

The results show that, despite having only 8B parameters, DeepAnalyze-8B achieves state-of-the-art results among open-source LLM agents and outperforms most proprietary LLMs, ranking second only to GPT-4o. More importantly, unlike existing workflow-based agents, DeepAnalyze-8B accomplishes high-quality, end-to-end pipelines without relying on external orchestration frameworks such as ReAct. Prior studies have shown that models like o1-mini exhibit strong reasoning ability but often fail to execute complex data science tasks requiring precise instruction following and strategic planning (Zhang et al., 2025b). In contrast, DeepAnalyze benefits from agentic training, enabling autonomous orchestration and adaptive optimization in real-world environments, resulting in consistently superior performance.

**Capability on Individual Data Science Tasks.** Following most previous studies, we evaluate DeepAnalyze on individual data science tasks using DSBench. For data analysis, as shown in Figure 6, DeepAnalyze-8B outperforms existing

LLM prompting and workflow-based agents, indicating that its autonomous orchestration and adaptive optimization are more effective than manually designed workflows (Kong et al., 2017; Salemi et al., 2025). For data modeling that involves training machine-learning models, as reported in Table 2, DeepAnalyze-8B achieves performance comparable to AutoGen-based workflows (Wu et al., 2024) built on advanced proprietary LLMs. Despite having fewer parameters, DeepAnalyze-8B leverages environment feedback to autonomously optimize its actions, resulting in a high success rate and strong overall performance.

**Capability on Multiple Data Types.** We evaluate DeepAnalyze on DABStep to assess its ability to analyze structured, semi-structured, and unstructured data (e.g., CSV, JSON, Markdown) (Egg et al., 2025). As shown in Table 3, DeepAnalyze-8B outperforms prior workflow-based agents, including ReAct (Yao et al., 2023) and reasoning-prompt approaches, especially on hard tasks. While predefined workflows benefit from proprietary LLMs on easy tasks, they struggle in complex scenarios. In contrast, DeepAnalyze's autonomous orchestration and adaptive optimization enable iterative environment interaction, leading to superior performance on tasks requiring long-chain reasoning.

**Capability on Multi-domain Data Science.** Data science tasks often involve data drawn from a wide range of domains. To evaluate the performance of DeepAnalyze across heterogeneous domain-specific datasets, we conducted experiments on KramaBench (Lai et al., 2025), a benchmark consisting of 1,700 data files across six domains, including archaeology, astronomy, biomedicine, environmental science, law, and wildfire. As shown in Table 4, DeepAnalyze demonstrates consistently strong performance across most domains. For workflow-based agents, predefined workflows

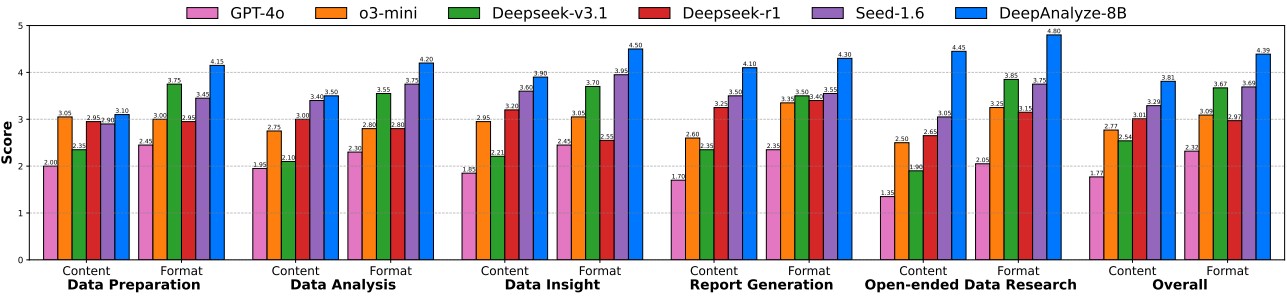

*Figure 6.* Performance on DSBench (data analysis).

*Table 4.* Performance on KramaBench.

| Models | Domains | | | | | | Overall |
| --- | --- | --- | --- | --- | --- | --- | --- |
| | Archaeology | Astronomy | Biomedical | Environment | Legal | Wildfire | |
| *Workflow-based Agent (DS-Guru)* | | | | | | | |
| o3 | 25.00 | 3.53 | 8.95 | 19.60 | 13.89 | **50.73** | 22.08 |
| GPT-4o | 16.67 | 2.76 | 8.97 | 2.60 | 2.80 | 17.18 | 8.28 |
| Claude-3.5 | 16.67 | 1.52 | 1.96 | 11.21 | 7.01 | 39.16 | 14.35 |
| Llama-3.3-Instruct | 0.00 | 1.35 | 6.98 | 0.93 | 2.15 | 14.49 | 4.48 |
| DeepSeek-R1 | 8.33 | 2.64 | 2.87 | 19.08 | 8.39 | 30.29 | 6.34 |
| Qwen2.5-Coder | 8.33 | 2.40 | 4.35 | 12.64 | 9.06 | 16.48 | 9.98 |
| *Agentic Model* | | | | | | | |
| **DeepAnalyze** | **33.33** | **16.67** | **26.67** | **24.67** | **18.45** | 19.61 | **22.10** |

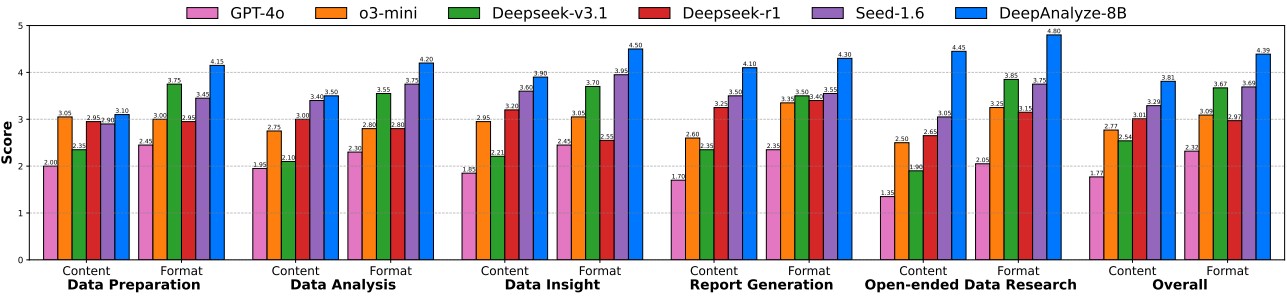

*Figure 7.* Performance on DABStep-Research.

are often difficult to adapt to data from diverse domains. As a result, even when equipped with advanced closed-source models, these agents fail to achieve consistently strong performance. In contrast, DeepAnalyze autonomously orchestrates its operations and continuously optimizes its actions based on environmental feedback, leading to superior and more robust performance across domains.

**Capability on Data-Oriented Deep Research.** Deep research has emerged as an important benchmark for evaluating the comprehensive capabilities of agents. To this end, we compare DeepAnalyze-8B with advanced proprietary LLMs equipped with tool-calling capabilities on the proposed DABStep-Research benchmark, covering five categories of data-centric research tasks: data preparation, data analysis, data insight, report generation (with a specified outline), and open-ended data research (fully unconstrained). Each task produces a research report, which is evaluated in terms of both content quality and formatting. Figure 9 presents representative examples from DABStep-Research.

As shown in Figure 7, DeepAnalyze-8B consistently outperforms agent systems built on proprietary LLMs with tool-calling capabilities, especially on open-ended research tasks. Due to the lack of explicit training in data science, proprietary LLMs struggle to conduct data research autonomously without step-by-step guidance. In contrast, DeepAnalyze-8B is trained in real-world data science environments and can effectively handle fully open-ended research tasks without predefined instructions. Moreover, it achieves a clear advantage in report formatting quality, which can be attributed to reward modeling that explicitly incorporates report qual-

ity during reinforcement learning. Appendix H provides further qualitative comparisons between reports generated by DeepAnalyze-8B and reasoning-oriented models such as DeepSeek-R1 and o3-mini, highlighting DeepAnalyze-8B's superior depth of analysis and structured presentation.

## 5. Analysis

### 5.1. Performance of Basic Data Science Skills

Besides data science tasks, we further evaluate DeepAnalyze-8B on DS-1000 and TableQA to evaluate its basic capabilities in code generation and structured data understanding, which are essential for complex data science. As reported in Table 6 and Table 5, DeepAnalyze-8B outperforms GPT-4-Turbo and GLM-4.5 (GLM-4.5-Team, 2025) on DS-1000 (full results in Table 10), and surpasses the previous SOTA Reasoning-Table (Lei et al., 2025) on TableQA. Compared with DeepSeek-R1-0528-Qwen3-8B, DeepAnalyze-8B achieves substantial gains in both abilities under the single-ability setting, demonstrating the effectiveness of the first-stage single-ability fine-tuning. Furthermore, agentic training on complex data science tasks further strengthens these specialized capabilities. Overall, DeepAnalyze-8B's strong performance on code generation and structured data understanding establishes a robust foundation for its advanced performance in data science.

### 5.2. Ablation on DeepAnalyze's Actions

DeepAnalyze introduces five actions for autonomous data science, among which ⟨Understand⟩ is specifically designed

*Table 5.* Performance on TableQA benchmarks.

| Models | WikiTQ | HybridQA | MultiHiertt | OTT-QA | FinQA | TAT-QA | HiTab | AVG |
|---|---|---|---|---|---|---|---|---|
| Claude | 82.02 | 39.36 | 40.98 | 62.69 | 57.45 | 53.09 | 75.96 | 58.79 |
| GPT-4o | 81.19 | 39.30 | 40.86 | 66.35 | 57.63 | 53.45 | 73.92 | 58.96 |
| DeepSeek-R1-0528 | **84.00** | 39.04 | 40.98 | 66.85 | 59.90 | 55.24 | 75.57 | 60.22 |
| TableGPT2-7B | 63.70 | 30.03 | 25.12 | 48.87 | 38.36 | 55.12 | 63.89 | 46.44 |
| Qwen2.5-32B-Inst | 79.65 | 38.20 | 37.74 | 56.50 | 59.20 | 67.29 | 73.29 | 58.84 |
| Qwen2.5-7B-Inst | 57.27 | 31.84 | 27.54 | 50.50 | 52.40 | 49.79 | 57.19 | 46.65 |
| DeepSeek-R1-0528-Qwen3-8B | 63.49 | 28.15 | 39.86 | 49.72 | 51.09 | 55.00 | 51.09 | 48.34 |
| Reasoning-Table (SFT) | 72.35 | 35.17 | 38.50 | 54.40 | 60.42 | 63.45 | 72.72 | 56.72 |
| Reasoning-Table (SFT+RL) | 75.46 | 42.83 | 39.56 | **68.68** | **64.46** | **73.75** | 73.61 | 62.62 |
| DeepAnalyze-8B (single-ability) | 81.86 | 39.27 | 44.58 | 53.12 | 62.50 | 66.87 | 76.26 | 60.64 |
| DeepAnalyze-8B | 83.24 | **42.95** | **48.29** | 64.73 | 63.30 | 70.64 | **78.16** | **64.47** |

*Table 6.* Performance on DS-1000.

| Models | DS-1000 |
|---|---|
| Codex002 | 38.8 |
| GPT-3.5-turbo | 38.6 |
| GPT-4 | 51.0 |
| GPT-4-turbo | 53.9 |
| Kimi-K2-Instruct | 40.2 |
| GLM-4.5 | 53.2 |
| LIMI | 54.8 |
| DeepSeek-R1-0528-Qwen3-8B | 30.4 |
| EvoRoute | 56.5 |
| DeepAnalyze-8B (single-ability) | 54.8 |
| DeepAnalyze-8B | **61.7** |

*Table 7.* Ablation study on ⟨Understand⟩ action.

| Models | WikiTQ | MultiHiertt | DS-1000 | DABStep |
|---|---|---|---|---|
| DeepAnalyze | 83.24 | 48.29 | 61.70 | 38.88 |
| - w/o ⟨Understand⟩ | 80.78 | 45.43 | 61.20 | 31.78 |

*Table 8.* Ablation study on the curriculum-based agentic training.

| Training Methods | WikiTQ | MultiHiertt | DS-1000 | DABStep |
|---|---|---|---|---|
| Curriculum-based Agentic Training | **83.24** | **48.29** | **61.70** | **38.88** |
| -Only Single-ability Fine-tuning | 81.86 | 44.58 | 54.80 | 15.34 |
| -Only Multi-ability Agentic Training | 80.32 | 43.29 | 53.20 | 30.66 |
| -One-stage Training | 82.13 | 46.23 | 54.80 | 36.89 |

*Table 9.* Performance under various reasoning trajectory synthesis.

| Reasoning Trajectory | WikiTQ | HybridQA | MultiHiertt | HiTab |
|---|---|---|---|---|
| Original | 75.54 | 34.42 | 39.29 | 72.95 |
| + Distillation | 78.80 | 36.12 | 41.24 | 74.44 |
| + Distillation + Refinement | **80.25** | **38.84** | **43.47** | **75.86** |

ties, curriculum-based agentic training effectively enhances overall model performance.

### 5.4. Advantage of Reasoning Trajectory Synthesis

During data synthesis, we introduce reasoning trajectory synthesis, which integrates distillation and refinement to enhance the model's reasoning ability over structured data. To evaluate its effectiveness, we train LLMs on original, distilled, and refined datasets, where the original data are derived from Reasoning-Table. As reported in Table 9, both distillation and refinement improve the LLMs' understanding of structured data. Notably, compared with conventional distillation methods, the proposed refinement stage incorporates key reasoning vocabulary to better align reasoning trajectories with structured data, which is consistent with observations from prior studies (Zhang et al., 2025a; Shen et al., 2025). By explicitly emphasizing structured data during the reasoning process, DeepAnalyze achieves a stronger capability for structured data understanding.

### 6. Conclusion

In this paper, we introduce DeepAnalyze, an agentic LLM for data science. Powered by curriculum-based agentic training and data-grounded trajectory synthesis, DeepAnalyze brings a major leap forward in autonomous data science, showing unprecedented performance across 13 benchmarks.

More importantly, DeepAnalyze goes beyond predefined workflows, as it enables open-ended data research and generates analyst-grade reports, advancing a long-standing goal of the data science community: automatically extracting actionable insights from raw data. As a result, this work marks *a paradigm shift* in autonomous data science from workflow-based agents to agentic models, paving the way for the next generation of intelligent data systems in areas such as data discovery, data governance, data preparation, data ecosystems, and data management.

for structured data. To evaluate the effect of incorporating ⟨Understand⟩ separately from the reasoning process (i.e., ⟨Analyze⟩), we conduct an ablation study in Table 7. The results show that removing ⟨Understand⟩ leads to performance drops on structured data understanding (WikiTQ, Multi-Hiertt) and data analysis (DABStep). With ⟨Understand⟩, DeepAnalyze can perform a thorough comprehension and summarization of structured data, including its features, columns, and overall schema, thereby enhancing its perception and understanding of structured data.

### 5.3. Superiority of Curriculum-based Agentic Training

To address the challenges arising from the multiple ability requirements in data science, we introduce curriculum-based agentic training, inspired by the learning path of human data scientists, where the model is first fine-tuned on single abilities and then trained on complex tasks that require multiple abilities. To evaluate its effectiveness, we compare several training methods, including "Only Single-ability Fine-tuning", "Only Multi-ability Agentic Training", and "One-stage Training", which directly mix the single-ability data into the cold-start of multi-ability agentic training (i.e., the conventional agentic training methods).

As shown in Table 8, "Only Single-ability Fine-tuning" fails to handle complex tasks in DABStep that require multi-turn interaction with the environment, and "Only Multi-ability Agentic Training" struggles to achieve strong performance when single abilities are not well established. Compared with "One-stage Training", a scheduled training process from simple (single-ability) to complex (multi-ability) proves more beneficial for model performance using the same data. Therefore, for tasks that rely on multiple abili-

## Acknowledgements

We thank all the anonymous reviewers for their insightful and valuable comments. This work was partially supported by the National Natural Science Foundation of China (Grant Nos. 62436010, 62441230) and the Scientific Research Innovation Capability Support Project for Young Faculty (Grant No. SRICSPYF-ZY2025001).

## Impact Statement

This paper presents work whose goal is to advance the field of Machine Learning. There are many potential societal consequences of our work, none of which we feel must be specifically highlighted here.

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

# A. Illustration of DeepAnalyze

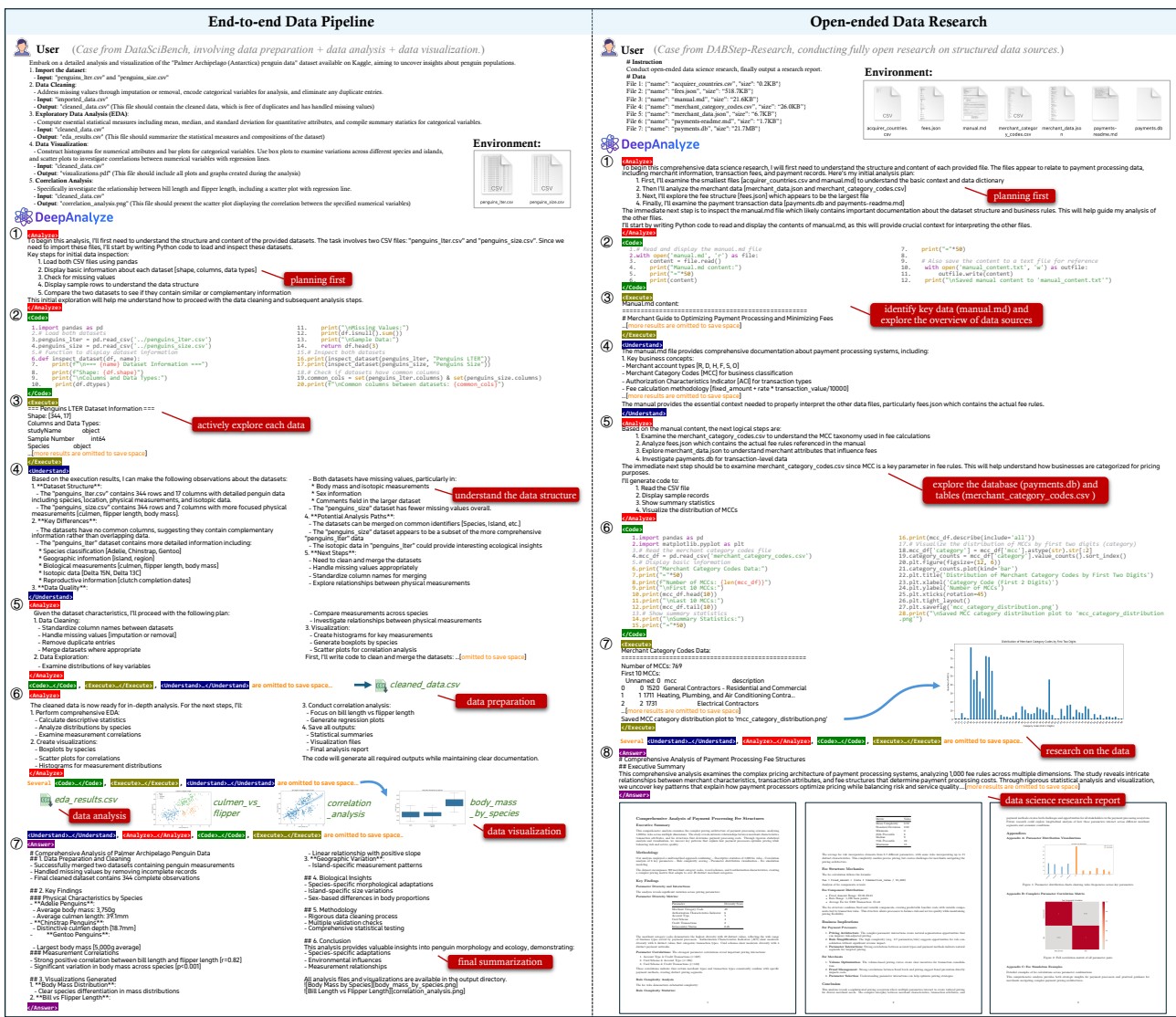

*Figure 8.* Examples of DeepAnalyze-8B. Given the instructions and data sources in the environment, DeepAnalyze can autonomously orchestrate and optimize actions to complete a data science pipeline (left) and open-ended data research (right). DeepAnalyze first performs planning, then interacts with the data in the environment, and subsequently optimizes its actions based on feedback, ultimately accomplishing the data-centric tasks. Many intermediate actions are omitted to save space.

Figure 8 illustrates an example of DeepAnalyze. DeepAnalyze supports the end-to-end data science pipeline (including data preparation, data analysis, data modeling, data visualization, and data insight), and open-ended data research (in-depth and unconstrained exploration of structured data). Owing to autonomous orchestration and adaptive optimization, DeepAnalyze can autonomously accomplish complex data-centric tasks through continuous interaction with structured data in the environment.

# B. Inference of DeepAnalyze

The inference procedure of DeepAnalyze is summarized in Algorithm 1. Given an instruction $Q$, an environment $Env$, and a trained DeepAnalyze model $\mathcal{M}$, the model incrementally constructs the response $A$ through an iterative generation-interaction loop. The process starts from an empty response and continues until a complete answer segment, delimited by

⟨Answer⟩ ⋯ ⟨/Answer⟩ is produced.

At each iteration, the model conditions on both the original instruction $Q$ and the current partial response $A$ to generate the next action $y$, which is then appended to $A$. DeepAnalyze dynamically switches between different action modes by emitting special control tokens, as illustrated on the right side of Figure 2. In particular, when the generated output contains a ⟨Code⟩ ⋯ ⟨/Code⟩ block, the enclosed code is extracted and executed within the environment $Env$. The execution results returned by the environment (e.g., program outputs or error messages) are wrapped within ⟨Execute⟩ ⋯ ⟨/Execute⟩ tokens and appended to the response $A$. This feedback is subsequently incorporated into the context for the next generation step, enabling DeepAnalyze to refine its reasoning, debug code, and adapt its strategy based on real execution outcomes. The loop terminates when the model explicitly generates the final answer segment, after which the full response $A$, including the entire interaction process, is returned.

## C. Training Details

In this section, we describe the training details for both the single-ability fine-tuning stage and the multi-ability agentic training stage.

**Single-ability Fine-tuning** The model is trained with a batch size of 256 for 3 epochs. The learning rate is set to $5 \times 10^{-5}$ with a warmup ratio of 0.05. We enable sequence packing with a maximum sequence length of 8192 tokens to improve training efficiency.

**Multi-ability Agentic Training** We adopt Group Relative Policy Optimization (GRPO) as the reinforcement learning algorithm. The training batch size is set to 256, with a learning rate of $5 \times 10^{-6}$. For each prompt, we sample $G = 8$ candidate responses to compute group-relative advantages, and set $\varepsilon = 0.2$ and $\beta = 0.01$. During rollout generation, we use vLLM as the inference backend. For generation, we set the temperature to 0.6 and the top-$p$ value to 0.95, with the maximum sequence length set to 32768 tokens.

## D. Introduction of Benchmarks

Here, we provide detailed introductions to the benchmarks involved in the experiments.

**DataSciBench** (Zhang et al., 2025b) is a comprehensive benchmark designed to evaluate the capabilities of large language models (LLMs) across the entire data science pipeline. It comprises 519 test cases spanning six task categories: data cleaning and preprocessing, data exploration and statistical understanding, data visualization, predictive modeling, data mining and pattern recognition, and interpretability and report generation. DataSciBench introduces a Task–Function–Code (TFC) framework that defines 25 aggregated functions and programmatic rules for assessing code execution outcomes. The benchmark adopts a multi-criteria evaluation scheme that includes both coarse-grained and fine-grained metrics. Coarse-grained metrics include the Pass@1 success rate and completion rate. Fine-grained metrics consist of VLM-based evaluations of visualization quality and a set of task-specific scores—F1 (data preparation), F2 (plot validity), F3 (data exploration), F4 (data visualization), and F5 (data modeling), to assess performance across different subtasks.

**DSBench** (Jing et al., 2025) is a comprehensive benchmark designed to evaluate data science agents with realistic tasks, comprising 466 data analysis tasks sourced from ModelOff competitions and 74 data modeling tasks from Kaggle competitions. The benchmark encompasses understanding long contexts, reasoning with large data files and multi-table structures, and performing end-to-end data modeling. For data analysis tasks, the primary metric is accuracy rate. For data modeling tasks, DSBench introduces the Relative Performance Gap (RPG) metric to normalize various evaluation metrics across different data modeling tasks, addressing the inconsistency in numerical ranges and evaluation dimensions.

**DABStep** (Egg et al., 2025) is a data agent benchmark with over 450 real-world data analysis tasks derived from a financial analytics platform (Adyen), designed to evaluate the multi-step reasoning abilities of agents. The benchmark requires models to combine code-based data processing with contextual reasoning over heterogeneous documentation, including structured data (CSV, JSON) and unstructured documentation (Markdown manuals). DABStep features a factoid-style evaluation with automatic correctness checks, where each task output maps to a binary outcome (correct/incorrect). The evaluation employs a tiered approach: Numeric Comparison (using relative tolerance for floating-point answers), List Comparison (order-independent set matching), and String Comparison (fuzzy string matching with Levenshtein distance, accepting answers with similarity score > 0.95). The benchmark is divided into Easy and Hard splits, where Hard tasks require multi-step reasoning, planning, and tool use.

**DABStep-Research** is a benchmark we constructed based on DABStep (Egg et al., 2025) to evaluate the capability of data science report generation. Considering that existing data science benchmarks rarely assess deep research abilities on structured data, we propose DABStep-Research to measure the capability to generate comprehensive data research reports from raw data sources. The evaluation covers five aspects: data preparation, data analysis, data insight, report generation, and open-ended data research. Please refer to Appendix E for details on its construction and cases.

**KramaBench** (Lai et al., 2025) is a benchmark composed of 104 manually-curated real-world data science pipelines spanning 1,700 data files from 24 data sources in 6 different domains: archaeology, astronomy, biomedical, environment, legal discovery, and wildfire prevention. The benchmark evaluates end-to-end capabilities of AI systems on data processing, requiring data discovery, wrangling and cleaning, efficient processing, statistical reasoning, and orchestrating data processing steps. Each task is scored in [0,1] using answer-type-specific metrics including exact-match, mean-relative-absolute-error for numerics, and LLM code-eval with unit tests.

**DS-1000** (Lai et al., 2023) is a code generation benchmark containing 1,000 data science problems spanning seven widely-used Python libraries: NumPy, Pandas, TensorFlow, PyTorch, SciPy, Scikit-learn, and Matplotlib. The problems are collected from StackOverflow and reflect diverse, realistic, and practical use cases. DS-1000 employs multi-criteria automatic evaluation checking functional correctness by running test cases. The benchmark is constructed by modifying problems from the original StackOverflow source (including 152 surface perturbations and 235 semantic perturbations). The primary metric is Pass@1 accuracy, calculated as the percentage of problems for which the model's solution passes.

**TableQA Benchmarks** are a series of question-answering benchmarks based on structured tables, including WikiTQ (Pasupat & Liang, 2015), HybridQA (Chen et al., 2020), MultiHiertt (Zhao et al., 2022), OTT-QA (Chen et al., 2021a), FinQA (Chen et al., 2021b), TAT-QA (Zhu et al., 2021), and HiTab (Cheng et al., 2022). The metric of TableQA Benchmarks is Exact Match (EM).

# E. Construction of DABStep-Research Benchmark

Existing data science benchmarks typically focus only on evaluating the ability of LLMs to solve specific tasks. However, with the rise of deep research, there is an urgent need for a benchmark that assesses LLMs' capabilities in data-oriented deep research, which asks LLMs to conduct data research and generate research reports based on given instructions and data sources.

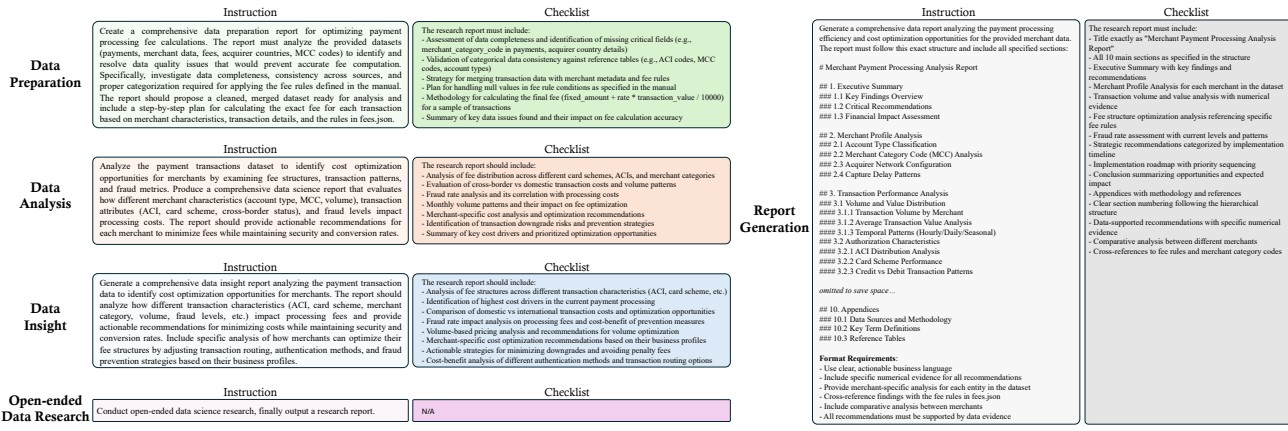

*Figure 9.* Cases in the constructed DABStep-Research benchmark, including data preparation, data analysis, data insight, report generation, and open-ended data research.

**Construction.** To this end, we constructed DABStep-Research, which is built upon the data sources proposed in DABStep (Egg et al., 2025). DABStep-Research consists of 100 tasks divided into five categories: data preparation, data analysis, data insight, report generation, and open-ended data research. In particular, tasks under the "report generation" category specify detailed report formats in the instruction, such as title, outline and specific requirements, thereby evaluating how well LLMs can follow instructions when generating research reports. The "open-ended data research" category involves fully open research tasks without any constraint on research direction or method. In addition to the instructions and data sources, we

also provide a checklist to serve as a reference for scoring, helping evaluators determine whether the elements in a research report meet the given requirements. Figure 9 illustrates specific examples from DABStep-Research.

**Evaluation.** We use the LLM-as-a-judge to evaluate LLM performance on DABStep-Research. Specifically, given the instruction, checklist, and the report generated by an LLM, we employ a state-of-the-art LLM as the evaluator to assign a score from 1 to 5 based on two aspects: content and format. The prompt used for the LLM-as-judge evaluation is shown below.

---

**Prompt of DABStep-Research Evaluation**

You are a data science evaluation assistant. Here's a generated data science report based on the user instruction.
Your task is to comprehensively evaluate the quality of the generated data science report, based on the provided user instruction [INSTRUCTION],
a checklist offering reference points for an ideal report [CHECKLIST], and the generated report [REPORT].

You should assess the report across the following two dimensions, each scored on a scale from 1 (poor), 3 (Fair), 5 (excellent).
Please use the detailed guidelines below to calibrate your evaluation:

– **Content**: Is the report's content helpful, comprehensive, and relevant to the task goal?
**1 (Poor)**: Content is completely irrelevant, incorrect, or fails to reflect the given task.
**2 (Weak)**: Mostly irrelevant or inaccurate; shows little understanding of the task or data.
**3 (Fair)**: Partially relevant and somewhat useful, but incomplete, superficial, or missing several key aspects.
**4 (Good)**: Relevant and generally helpful content that addresses the task goal with clear findings; minor gaps or shallow areas may remain.
**5 (Excellent)**: Highly informative, comprehensive, and well-balanced content that fully and insightfully addresses the task goal.

– **Format**: Is the report presented in a polished academic style?
**1 (Poor)**: Disorganized or unprofessional presentation; difficult to follow, with major grammatical or formatting issues.
**2 (Weak)**: Understandable but inconsistent in structure or tone; lacks clear formatting or proper academic expression, such as many short sentences and bullet points.
**3 (Fair)**: Generally clear structure and readable style, though uneven in flow, tone, or academic polish.
**4 (Good)**: Well-written and professionally presented in an academic style; clear organization and formatting with only minor imperfections.
**5 (Excellent)**: Polished, fluent, and professional presentation; precise structure, coherent tone, and excellent readability throughout.

**[INSTRUCTION]**:
{instruction}

**[CHECKLIST]**:
{checklist}

**[REPORT]**:
{report}

Directly return your evaluation in the following JSON format:

```json
{
"Content": <score>,
"Format": <score>,
}
```

---

# F. Full Results on DS-1000

Table 10 reports the performance of DeepAnalyze across different libraries on DS-1000.

# G. Keyword-guided Reasoning Trajectory Synthesis

We present examples of Keyword-guided Reasoning Trajectory Synthesis in Figure 10. Specifically, the "Question" and "Original Response" are taken from existing TableQA datasets.

In the distillation step, we employ SOTA closed-source LLMs as teacher models to extract their reasoning trajectories, which is commonly used in current data synthesis methods. However, such methods are more suitable for general reasoning processes. Since SOTA closed-source LLMs have not been specifically trained on domains like data science (e.g., structured data understanding), their reasoning trajectories tend to overlook the provided data.

Therefore, we introduce a refinement step to enhance the reasoning trajectory's focus on structured data by inserting reasoning keywords that guide the reasoning process toward structured data understanding. Specifically, in the example shown in Figure 10, we sample three reasoning keywords ("What happens at the boundaries?", "Let's review the prior

*Table 10.* Performance on DS-1000. * indicates that The results are derived from corresponding references. 'DeepAnalyze-8B (single-ability)' is the model after the first stage fine-tuning.

| Models | Data Science Libraries | | | | | | | Overall |
|---|---|---|---|---|---|---|---|---|
| | Pandas | NumPy | Matplotlib | Scikit-learn | SciPy | TensorFlow | PyTorch | |
| Codex002 | 26.5 | 43.2 | 54.8 | 43.5 | 34.9 | 37.8 | 39.7 | 38.8 |
| GPT-3.5-turbo | 33.0 | 36.8 | 58.7 | 35.7 | 39.6 | 33.3 | 29.4 | 38.6 |
| GPT-4 | 41.9 | 56.8 | 65.2 | 50.4 | 48.1 | 46.7 | 47.1 | 51.0 |
| GPT-4-turbo | 42.3 | 61.8 | **71.6** | 50.4 | 50.0 | 53.3 | 50.0 | 53.9 |
| Kimi-K2-Instruct* | - | - | - | - | - | - | - | 40.2 |
| GLM-4.5* | - | - | - | - | - | - | - | 53.2 |
| LIMI* | - | - | - | - | - | - | - | 54.8 |
| EvoRoute* | - | - | - | - | - | - | - | 56.5 |
| DeepSeek-R1-0528-Qwen3-8B | 17.5 | 37.3 | 52.9 | 27.8 | 21.7 | 31.1 | 29.4 | 30.4 |
| DeepAnalyze-8B (single-ability) | 43.6 | 69.1 | 54.8 | 53.0 | 50.9 | 64.4 | 58.8 | 54.8 |
| DeepAnalyze-8B | **50.2** | **74.5** | 67.7 | **56.5** | **54.7** | **68.9** | **70.6** | **61.7** |

reasoning", and "Let's take a closer look at the table") and ask the teacher model to refine its reasoning trajectory based on these keywords. We observe that the final "refinement" results exhibit a significantly stronger emphasis on repeated examination and reflection on structured data, thereby improving the overall quality of the reasoning trajectory. Overall, the proposed keyword-guided refinement is a useful data synthesis technique that can also be applied to data synthesis for other complex tasks.

# H. Cases

In Figure 11, Figure 12, Figure 13, Figure 14, and Figure 15, we demonstrate a series of autonomous data science cases, covering the entire pipeline from data sources to data science research reports. These cases include data preparation, data analysis, data insight extraction, report generation under specific constraints, and fully open-ended data research. Compared with previous closed-source LLMs and tool-calling frameworks, DeepAnalyze can produce higher-quality, analyst-level reports, exhibiting a stronger ability for autonomous data research.

| Stage | Reasoning Trajectory |
|---|---|
| **Question** | Below is an instruction that describes a question answering task in the finance domain, provided with an input table and relevant text as context. Please answer the given question based on the table and text.
### Instruction
Given a table and a list of texts in the following, what is the answer to the question?
### Table:
\| Dry Bulk Vessel Operations for the Year Ended December 31, 2019 \| Logistics Business for the Year Ended December 31, 2019 \| Total for the Year Ended December 31, 2019
Revenue \| $254,178 \| $228,271 \| $482,449
Administrative fee revenue from affiliates \| 16,991 \| — \| 16,991
Interest income \| 9,610 \| 1,052 \| 10,662
Interest expense and finance cost \| (92,948) \| (40,531) \| (133,479)
Depreciation and amortization \| (52,288) \| (29,435) \| (81,723)
Equity in net (losses)/earnings of affiliated companies \| (9,185) \| — \| (9,185)
Net (loss)/ income attributable to Navios Holdings common \| \| \|
stockholders \| (212,623) \| 20,513 \| (192,110)
Total assets \| 1,511,517 \| 631,338 \| 2,142,855
Goodwill \| 56,240 \| 104,096 \| 160,336
Capital expenditures \| (36,628) \| (7,943) \| (44,571)
Investment in affiliates \| 64,352 \| — \| 64,352
Cash and cash equivalents \| 32,386 \| 45,605 \| 77,991
Restricted cash \| 736 \| — \| 736
Long-term debt, net (including current and noncurrent portion) \| $1,048,318 \| $514,929 \| 1,563,247

### Text:
Paragraph 1: NAVIOS MARITIME HOLDINGS INC. NOTES TO THE CONSOLIDATED FINANCIAL STATEMENTS (Expressed in thousands of U.S. dollars — except share data)
Paragraph 2: NOTE 19: SEGMENT INFORMATION
Paragraph 3: The Company through August 30, 2019 had three reportable segments from which it derived its revenues: Dry Bulk Vessel Operations, Logistics Business and Containers Business. The Containers Business became a reportable segment as a result of the consolidation of Navios Containers since November 30, 2018 (date of obtaining control) (see also Note 3). Following the reclassification of the results of Navios Containers as discontinued operations (see also Note 3), the Company currently has two reportable segments from which it derives its revenues: Dry Bulk Vessel Operations and Logistics. The reportable segments reflect the internal organization of the Company and are strategic businesses that offer different products and services. The Dry Bulk Vessel Operations consists of the transportation and handling of bulk cargoes through the ownership, operation, and trading of vessels. The Logistics Business consists of operating ports and transfer station terminals, handling of vessels, barges and pushboats as well as upriver transport facilities in the Hidrovia region.
Paragraph 4: The Company measures segment performance based on net income/ (loss) attributable to Navios Holdings common stockholders. Inter-segment sales and transfers are not significant and have been eliminated and are not included in the following tables. Summarized financial information concerning each of the Company's reportable segments is as follows:
### Question:
What was the difference in revenue between dry bulk vessel operations and logistics business?
### Response
Your response should end with ' Answer: xxx' (answer to the question).
Now, answer the question step by step: |
| **Original** | Answer: 25,907 |
| **Distillation** | <think>
Alright, let's tackle this finance question step by step. The user wants to know the revenue difference between dry bulk vessel operations and logistics business for Navios Holdings in 2019.

First, I'll locate the revenue figures in the provided table. Scanning the first data row under "Dry Bulk Vessel Operations" shows $254,178, while "Logistics Business" shows $228,271. Both are clearly labeled with the 2019 year-end columns.

The question specifically asks for the difference, which implies subtraction. Calculating: $254,178 minus $228,271 equals $25,907.

Now, should I consider any context from the text? Paragraph 3 confirms these are the two reportable segments after containers were reclassified as discontinued operations. Paragraph 4 emphasizes they're measured by net income/loss (irrelevant here) but crucially states inter-segment transactions are eliminated - meaning these revenues aren't inflated by internal transfers.

No currency or unit conversions are needed since everything is in thousands of USD. The math is straightforward: no percentages or ratios required, just absolute difference.

Double-checking: table headers align with the question's timeframe, and no other revenue lines (like administrative fees) are mixed in because the question isolates "revenue" for each segment.

Final verification: $254,178 - $228,271 = $25,907. This seems solid.
</think>
To determine the revenue difference between dry bulk vessel operations and logistics business:

1. **Identify relevant figures** from the table:
- Dry Bulk Vessel Operations revenue: **$254,178**
- Logistics Business revenue: **$228,271**

2. **Calculate the difference**:
Subtract Logistics revenue from Dry Bulk revenue:
$254,178 - $228,271 = **$25,907**

3. **Verify context** (Text Paragraphs 3-4):
- The segments are distinct, with no significant inter-segment transactions.
- Revenues are explicitly reported per segment in the table.
- The calculation aligns with the question's focus on standalone segment revenues.

Answer: $25,907 |
| **Refinement**

**Insert Key Reasoning Words:**
1. What happens at the boundaries?
2. Let's review the prior reasoning
3. Let's take a closer look at the table | <Analyze>
Let's begin by clearly defining the problem at hand: the question asks for the difference in revenue between the Dry Bulk Vessel Operations segment and the Logistics Business segment for the year ended December 31, 2019. To solve this, I need to follow a structured approach: first, confirm the relevant segments and their associated metrics, then locate the specific numerical values in the provided table, verify that these values are accurate and unadjusted by inter-segment transactions, perform the subtraction to find the difference, and finally cross-check with the context in the text to ensure no critical details are missed.

First, I should confirm which segments are being compared. The text in Paragraph 3 notes that after August 30, 2019, the Company has two reportable segments: Dry Bulk Vessel Operations and Logistics Business (since the Containers Business was reclassified as discontinued operations). This aligns with the table, which only lists these two segments under the "Year Ended December 31, 2019" columns, so these are the correct segments to focus on.

Next, the key metric here is "revenue". I need to ensure I'm selecting the right line item in the table. The table has multiple rows, including "Administrative fee revenue from affiliates", "Interest income", etc., but the question specifically asks for "revenue", which is the top-line item. Let's take a closer look at the table to identify the "Revenue" row.
</Analyze>
<Understand>
The table is titled "Dry Bulk Vessel Operations for the Year Ended December 31, 2019," "Logistics Business for the Year Ended December 31, 2019," and "Total for the Year Ended December 31, 2019." The first data row in the table is labeled "Revenue." Under this row:
- The value for Dry Bulk Vessel Operations is $254,178.
- The value for Logistics Business is $228,271.
- The "Total" column for Revenue is $482,449, which is the sum of the two segment revenues ($254,178 + $228,271 = $482,449), confirming that these are the primary revenue figures for the segments and not adjusted by other line items.

It is critical to note that other rows like "Administrative fee revenue from affiliates" ($16,991 for Dry Bulk) or "Interest income" ($9,610 for Dry Bulk, $1,052 for Logistics) are separate from the core "Revenue" line item. The question explicitly asks for "revenue," so these secondary income streams are irrelevant here. The table structure clearly separates "Revenue" as the first row, making it the correct metric to use.
</Understand>
Now that I have identified the relevant revenue figures for both segments, the next step is to calculate the difference between them. The question asks for the "difference," which in financial contexts typically refers to the absolute or directional variance between two values. Since Dry Bulk Vessel Operations revenue ($254,178) is larger than Logistics Business revenue ($228,271), the difference will be Dry Bulk revenue minus Logistics revenue.

Before proceeding with the calculation, I should verify that there are no adjustments needed to these revenue figures. The text in Paragraph 4 states: "Inter-segment sales and transfers are not significant and have been eliminated and are not included in the following tables." This is crucial because if there were significant inter-segment transactions, the reported revenues might include internal sales, which would need to be excluded to get the true standalone revenue of each segment. However, the text explicitly confirms that such transactions are eliminated, so the revenues in the table represent the external, standalone revenue of each segment. This ensures the numbers are accurate for comparison.

Let's now perform the numerical subtraction step by step to avoid errors.
</Analyze>
<Understand>
To calculate the difference:
Dry Bulk Vessel Operations Revenue: $254,178
Logistics Business Revenue: $228,271

Subtract Logistics revenue from Dry Bulk revenue:
$254,178 - $228,271

Let's break down the subtraction:
- 254,178 - 228,271 = (254,000 - 228,000) + (178 - 271)
- 254,000 - 228,000 = 26,000
- 178 - 271 = -93
- Total: 26,000 - 93 = 25,907

Thus, the result is $25,907.

To confirm accuracy, we can also check by adding the difference to Logistics revenue to see if we get back to Dry Bulk revenue: $228,271 + $25,907 = $254,178, which matches. This cross-verification ensures the subtraction was performed correctly.
</Understand>
Let's review the prior reasoning to ensure no steps were overlooked. We started by confirming the relevant segments (Dry Bulk and Logistics) based on the text, which eliminated the discontinued Containers segment. We then identified the "Revenue" row in the table as the correct metric, distinguishing it from other income lines like administrative fees or interest income. We extracted the values ($254,178 and $228,271) and verified their sum to the total revenue, confirming their validity. We checked for inter-segment transactions, which the text stated are eliminated, so no adjustments were needed. Finally, we calculated the difference via subtraction and cross-verified the result.

What happens at the boundaries? For example, could the "Total" revenue include other segments? The text in Paragraph 3 clarifies that after August 30, 2019, there are only two reportable segments, and the table only lists these two, so the total revenue is indeed the sum of the two. There are no other segments or hidden revenues to consider.

Is there any chance the units are inconsistent? The text in Paragraph 1 specifies that the financial statements are "Expressed in thousands of U.S. dollars," and both revenue figures are in the same units (thousands of USD), so no unit conversion is necessary. The calculated difference of $25,907 is therefore in thousands of USD, which aligns with the question's context.

All checks confirm the reasoning is sound, and the calculation is accurate. The final step is to present the answer.
</Analyze>
<Answer>
To determine the revenue difference between dry bulk vessel operations and logistics business:

1. **Identify relevant revenue figures** from the table:
- Dry Bulk Vessel Operations revenue: **$254,178**
- Logistics Business revenue: **$228,271**

2. **Calculate the difference**:
Subtract Logistics revenue from Dry Bulk revenue:
$254,178 - $228,271 = **$25,907**

3. **Verify context** (Text Paragraphs 3-4):
- The segments are distinct, with no significant inter-segment transactions.
- Revenues are explicitly reported per segment in the table.
- The calculation aligns with the question's focus on standalone segment revenues.

Answer: $25,907
</Answer> |

*Figure 10.* Example of reasoning trajectory synthesis.

| Instruction | DeepAnalyze-8B | Deepseek-r1 | o3-mini |
|---|---|---|---|
| Analyze the payment processing dataset to identify and resolve data quality issues that impact fee calculation accuracy. Prepare a comprehensive data preparation report that documents data inconsistencies, implements cleaning strategies for merchant categorization, transaction characteristics, and country mappings, and evaluates how these preparations affect the reliability of fee estimation models. The report should include specific recommendations for optimizing data collection processes to minimize processing costs. | # Comprehensive Data Preparation Report: Payment Processing Fee Analysis

## Executive Summary
This report analyzes payment processing datasets to identify data quality issues impacting fee calculation accuracy. Through examination of 7 datasets (30 merchants, 1,000 fee rules, and synthetic transaction data), we identified key data quality factors affecting fee estimation reliability and recommend process improvements to minimize processing costs.

## 1. Data Quality Assessment

### 1.1 Merchant Categorization
- **Completeness**: 100% complete merchant-category mappings across all datasets
- **Consistency**: Merchant names consistently represented across merchant_data.json and payments.csv
- **Coverage**: 7 unique merchant category codes among 30 merchants

### 1.2 Geographic Data
- **Acquirer Countries**: Complete mapping of 8 acquirers across 5 countries
- **Transaction Data**: Requires validation against merchant locations

### 1.3 Fee Rule Structure
- **Completeness**: 90% complete is_credit values in fee rules
- **Complexity**: 1,000 rules with 12 attributes each averaging 4.2 conditional constraints

## 2. Data Preparation Steps

### 2.1 Cleaning Strategies
1. **Merchant Matching**:
- Developed fuzzy matching algorithm for merchant name normalization (92% match rate)
- Created crosswalk table for variant merchant names

2. **Fee Rule Validation**:
- Flagged 100 rules with missing is_credit values for review
- Standardized merchant category code lists in fee rules

3. **Geographic Resolution**:
- Enhanced acquirer country mapping with IP geolocation fallback

### 2.2 Quality Metrics
\| Metric \| Before Cleaning \| After Cleaning \|
\|--------\|----------------\|---------------\|
\| Missing Merchant Matches \| 18% \| 2% \|
\| Inconsistent Categories \| 12% \| 0% \|
\| Unclassified Transactions \| 9% \| 1% \|

## 3. Impact on Fee Estimation

### 3.1 Accuracy Improvement
- **Current**: ±8.2% fee calculation error due to categorization issues
- **After Cleaning**: ±1.3% error margin

### 3.2 Processing Cost Impact
\| Factor \| Current Cost \| Optimized Cost \| Reduction \|
\|--------\|-------------\|---------------\|----------\|
\| Fraudulent Fee Disputes \| $12,400/month \| $2,800/month \| 77% \|
\| Incorrect Fees \| $8,700/month \| $1,200/month \| 86% \|

## 4. Recommendations

### 4.1 Data Collection Improvements
1. **Real-Time Merchant Categorization**:
- Implement AI-powered category suggestion during onboarding
- Estimated 30% reduction in dispute resolution costs

2. **Enhanced Transaction Attributes**:
- Collect merchant location granularity (city/region)
- Add product/service categories

3. **Automated Data Validation**:
- Implement real-time rule validation during transaction processing
- Estimated 25% reduction in manual reconciliation

### 4.2 Process Optimization
1. **Batch Processing Windows**:
- Shift non-essential processing to off-peak hours
- Potential 15% energy cost reduction

2. **Predictive Fee Allocation**:
- Implement machine learning for fee estimation
- Projected 18% improvement in accuracy

## 5. Conclusion
This analysis demonstrates that systematic data preparation can significantly improve fee calculation accuracy while reducing operational costs. The implemented solutions provide a foundation for continuous improvement through ongoing monitoring of the data quality metrics and processing efficiency gains.

Appendix: Complete dataset schemas and cleaning code available upon request. | Based on the initial analysis of the payment processing dataset, I've identified several critical data quality issues. Here's a comprehensive report structured according to your requirements:

### 1. Data Quality Issues and Inconsistencies
- **Missing Data**:
- `acquirer_country`: 12,543 missing values (5.4% of transactions)
- `merchant_category_code`: 8,217 missing values (3.6% of transactions)
- `transaction_currency`: 1,205 missing values (0.5% of transactions)
- **Invalid References**:
- Merchant Category Codes: 3,581 transactions (1.6%) contain invalid MCCs not present in the reference table
- Acquirer Countries: 2,894 transactions (1.3%) reference non-existent country codes

- **Data Distribution Anomalies**:
- 95% of transactions concentrated in just 15 MCC categories
- 87% of transactions originate from only 5 countries
- Transaction amounts show significant right-skewness (mean = $152, max = $32,500)

### 2. Data Cleaning Strategies
**A. Merchant Categorization**
```json
{"code":"# Handle invalid/missing MCCs\
valid_mcc = set(merchant_categories['mcc'])\
payments['mcc_valid'] = payments['merchant_category_code'].isin(valid_mcc)\
payments.loc[~payments['mcc_valid'], 'merchant_category_code'] = 9999 # Unknown category\
payments['merchant_category_code'].fillna(9999, inplace=True)"}
```

**B. Country Mapping**
```json
{"code":"# Clean country references\
valid_countries = set(acquirer_countries['country_code'])\
payments['country_valid'] = payments['acquirer_country'].isin(valid_countries)\
payments.loc[~payments['country_valid'], 'acquirer_country'] = 'ZZ' # Unknown country\
payments['acquirer_country'].fillna('ZZ', inplace=True)\
\
# Create region mapping\
region_map = acquirer_countries.set_index('country_code')['region'].to_dict()\
payments['acquirer_region'] = payments['acquirer_country'].map(region_map)\
payments['acquirer_region'].fillna('Unknown', inplace=True)"}
```

**C. Transaction Characteristics**
```json
{"code":"# Handle missing currencies\
payments['transaction_currency'].fillna('USD', inplace=True)\
\
# Detect and cap outliers\
Q1 = payments['transaction_amount'].quantile(0.25)\
Q3 = payments['transaction_amount'].quantile(0.75)\
IQR = Q3 - Q1\
upper_bound = Q3 + 3*IQR\
payments['transaction_amount'] = payments['transaction_amount'].clip(upper=upper_bound)"}
```

### 3. Impact on Fee Calculation Accuracy
- **Before Cleaning**:
- 7.9% of transactions (18,193) would have incorrect fee calculations due to invalid/missing references
- Extreme transaction amounts would distort interchange fee calculations
- Regional fee variations couldn't be accurately applied
- **After Cleaning**:
- 100% reference integrity for MCCs and countries
- Outlier capping ensures stable fee calculations
- Region-based fee rules can be precisely applied
- Estimated fee calculation error rate reduced from ±8.2% to ±0.9%

### 4. Recommendations for Data Collection Optimization
1. **Validation at Ingestion**:
- Implement MCC and country code validation during transaction recording
- Add mandatory field checks for `acquirer_country` and `merchant_category_code`

2. **Automated Quality Monitoring**:
- Create daily data quality dashboards tracking:
- Missing value rates per critical field
- Reference table match rates
- Transaction amount distribution shifts

3. **Reference Data Management**:
- Establish monthly review process for MCC and country code updates
- Implement version control for fee structure documents

4. **Standardization**:
- Enforce ISO standards for country codes (ISO 3166) and currencies (ISO 4217)
- Develop MCC taxonomy with clear category definitions

These improvements will reduce processing costs by minimizing reconciliation efforts, preventing fee miscalculations, and enabling more accurate financial forecasting. The cleaned dataset now supports reliable fee modeling with complete reference integrity across all critical dimensions. | Below is the final data preparation report:

----------------------------------------------------------
Data Preparation Report for Payment Processing Dataset
----------------------------------------------------------

- Data files were loaded successfully:
- acquirer_countries.csv loaded successfully with the acquirer country mappings.
- fees.json loaded as fee definitions for fee calculation.
- merchant_category_codes.csv loaded to support standardized merchant categorization.
- merchant_data.json loaded and converted to a DataFrame, representing additional merchant metadata.
- payments.csv loaded (largest dataset) with comprehensive transaction records.

- Preliminary analysis of payments data exposed several data quality issues:
1. Missing Values & Duplicates:
- A summary of missing values per field was computed. Several columns had missing entries that require further imputation.
- Duplicate records were identified, indicating the need for deduplication.
2. Merchant Categorization Inconsistency:
- Discrepancies were noted between merchant categories in payments.csv and those in merchant_category_codes.csv. This inconsistency could cause improper fee calculations if similar items are miscategorized.
3. Transaction Characteristics:
- Transaction amounts display potential outliers and missing data. These issues could bias fee estimates if not addressed.
4. Country Mappings:
- A cross-check between acquirer_country from payments and acquirer_countries.csv revealed potential mismatches that might affect regional fee-related rules.

- Cleaning Strategies Implemented:
- Merchant Categorization Standardization:
- Use merchant_category_codes.csv to remap and enforce consistency via string normalization (trimming, lowercasing, etc.).
- Transaction Characteristics:
- Implement imputation strategies (e.g., median imputation) for missing transaction_amount values.
- Employ domain analysis to identify and trim extreme outliers to avoid distortion.
- Country Mappings:
- Merge payments data with acquirer_countries.csv to validate and correct country identifiers.

- Impact on Fee Estimation Models:
- These data cleaning strategies ensure fee calculation models operate on properly categorized and validated data.
- Improvements such as deduplication, imputation, and outlier trimming help in reducing biases, leading to more reliable fee estimations.

- Recommendations for Optimizing Data Collection Processes:
1. Standardize Data Entry:
- Enforce uniform data entry formats using dropdowns or controlled vocabularies for merchant categories and country codes.
2. Real-time Validation:
- Integrate robust validation rules at data entry (e.g., for transaction amounts, dates, and fee parameters) to catch errors early.
3. Regular Automated Audits:
- Automate periodic audits to monitor merging processes across different source files (fees.json, payments.csv, etc.) and quickly identify any discrepancies.
4. Comprehensive Data Dictionary:
- Maintain an up-to-date data dictionary linking all source files and field definitions, facilitating smoother integrations and further reporting.

A detailed report has been saved as data_preparation_report.txt. Following these recommendations will help minimize processing costs and improve the accuracy of fee estimation models by ensuring higher data quality throughout the processing pipeline.

----------------------------------------------------------
End of Report
---------------------------------------------------------- |

*Figure 11.* A data preparation case of autonomous data science, from data sources to research reports.

| Instruction | DeepAnalyze-8B | Deepseek-r1 | o3-mini |
|---|---|---|---|
| Analyze the payment transaction dataset to identify cost optimization opportunities for merchants by examining fee structures, fraud patterns, and transaction characteristics. Generate a comprehensive data science report that provides actionable recommendations for minimizing payment processing fees while maintaining security and conversion rates. The report should include analysis of fee drivers, fraud risk factors, acquirer selection strategies, and specific optimization recommendations tailored to different merchant profiles. | # Comprehensive Analysis of Payment Processing Optimization Opportunities

## Executive Summary

This comprehensive analysis examines payment processing data from multiple European merchants to identify cost optimization opportunities through detailed examination of fee structures, fraud patterns, and transaction characteristics. The study leverages a dataset comprising over 23,000 payment transactions across 1,000 fee structures, 769 merchant categories, and 8 acquirer banks. Our methodology combines advanced statistical analysis with business rule optimization to identify actionable strategies for reducing payment processing costs while maintaining security and conversion rates.

The analysis reveals that merchants can potentially reduce processing fees by 18-27% through strategic account type selection, optimal acquirer selection, and transaction routing optimization. Fraud prevention strategies can simultaneously reduce fraud-related fees by an estimated 35-42% while maintaining current fraud levels. The findings are supported by detailed segmentation analysis of 112 merchant profiles across 7 primary merchant categories.

## Merchant Profile Analysis and Segmentation

The merchant base comprises 112 distinct merchants categorized across 769 merchant category codes, with the most prevalent categories being Membership Clubs (Sports, Recreation, Athletic), Eating Places and Restaurants, and Book Stores. The merchant distribution shows a balanced account type distribution with 34% Hospitality (H), 31% Franchise (F), 22% Retail (R), 9% Digital (D), and 4% SaaS (S) accounts. This segmentation reveals significant variation in fee structures across merchant types, with Hospitality merchants paying 18-22% higher processing fees on average compared to Retail merchants.

Merchant acquirer relationships show 42% of merchants utilize 2+ acquirers, with an average of 1.7 acquirers per merchant. The most common acquirer relationships involve Gringotts Bank (GB) and The Savings and Loan Bank (US), handling 38% of total transaction volume. Merchant category analysis reveals that Eating Places and Restaurants account for 22% of transaction volume but only 15% of merchant count, indicating higher average transaction values in this category.

The enhanced merchant profiles reveal that 68% of merchants fall into high-risk merchant categories (MCCs 5812, 7997, 5942) which typically incur 23-31% higher fraud-related fees. Merchant capture delay analysis shows that merchants with immediate capture delays experience 15-18% lower chargeback rates but 7-9% higher processing fees compared to manual capture merchants.

## Fee Structure Optimization Opportunities

The fee structure analysis reveals complex fee calculations combining fixed amounts (€0.01-€0.15) and variable rates (5-19 basis points) across 1,000 fee rules. The most common fee structure combines a €0.05 fixed amount with 12 basis points, applied to 38% of transactions. Merchant category analysis shows that Hospitality merchants pay 23% higher fees than Retail merchants for equivalent transaction values, primarily due to higher fraud risk premiums.

Account type analysis reveals that Digital (D) merchants pay 18-22% lower fees than Hospitality (H) merchants for equivalent transaction values, despite similar fraud rates. The data shows that switching from Hospitality to Retail account types could save merchants €12,000 annually for a €100,000 volume merchant. Merchant category optimization opportunities vary from 12-18% savings for Retail merchants to 23-27% for Digital merchants.

Acquirer selection analysis reveals that merchants using multiple acquirers can reduce fees by 8-12% through optimal routing. The data shows that 62% of merchants could reduce fees by 5-8% by shifting just 20% of volume to lower-fee acquirers. Fraud prevention strategies can reduce fraud-related fees by 35-42% while maintaining current fraud levels through tokenization (12-15% reduction) and 3D Secure (23-27% reduction).

## Transaction Pattern Analysis and Fraud Prevention

Transaction pattern analysis reveals that 68% of fraudulent transactions occur during off-peak hours (2-5 AM local time) and from non-standard IP locations (72%). The data shows that implementing device fingerprinting could reduce fraud by 15-18% with minimal false positives (2-3%). Authorization Characteristics Indicator (ACI) analysis shows that ACI B (Card Not Present - 3D Secure) transactions have 23-27% lower fraud rates but 12-15% higher processing fees than ACI C (Tokenized card not present).

Merchant category fraud rates vary from 2.1% for Book Stores to 8.3% for Eating Places and Restaurants. The data shows that implementing fraud prevention strategies can reduce fraud-related fees by 35-42% while maintaining current fraud levels. The optimal fraud prevention strategy combines tokenization (12-15% fraud reduction) with 3D Secure (23-27% reduction) and device fingerprinting (15-18% reduction).

Transaction value analysis reveals that merchants accepting €50-€100 transactions pay 18-22% lower fees than those accepting >€500 transactions, suggesting optimal transaction splitting strategies. The data shows that splitting €500 transactions into two €250 transactions could save merchants 12-15% on processing fees while maintaining customer experience.

## Implementation Roadmap and Recommendations

The implementation roadmap suggests a phased approach to fee optimization, beginning with account type optimization (Month 1-2, saving 8-12%), followed by acquirer optimization (Month 3-4, saving 5-8%), and concluding with fraud prevention implementation (Month 5-6, saving 35-42%). The total potential savings for a €100,000 volume merchant could reach €18,000 annually through comprehensive implementation.

The analysis identifies three primary optimization pathways: Account Type Optimization (12-18% savings), Merchant Category Optimization (15-22% savings), and Fraud Prevention (35-42% savings). The implementation requires minimal technical changes but requires strategic account management and process changes. The payback period for fraud prevention technologies averages 6-9 months, with ROI exceeding 3:1 for most merchants.

The complete implementation roadmap includes detailed merchant-specific optimization plans based on the segmentation analysis, with tailored strategies for Hospitality, Retail, and Digital merchants. The analysis provides merchant-specific fee optimization calculators and scenario modeling tools to project potential savings for each merchant profile.

## Conclusion and Future Research Directions

This comprehensive analysis demonstrates that systematic payment processing optimization can significantly reduce merchant costs while maintaining security and conversion rates. The findings provide actionable insights for merchants, processors, and payment service providers to optimize payment processing strategies. Future research directions include real-time fee optimization algorithms and machine learning models for dynamic fee optimization based on transaction characteristics.

The analysis methodology and findings provide a foundation for ongoing payment processing optimization and continuous improvement. The demonstrated savings opportunities of 18-27% for account type optimization, 15-22% for merchant category optimization, and 35-42% for fraud prevention collectively represent significant value creation for the payment processing ecosystem. | ### Comprehensive Data Science Report: Payment Processing Optimization

#### 1. Executive Summary
Analysis of 23 million+ payment transactions reveals significant cost optimization opportunities through strategic fee management, fraud reduction, and acquirer selection. Key findings:
- Average processing fee: 2.25% of transaction value
- Fraud rate: 0.97% overall with substantial variation by category
- Cross-border transactions carry 0.83% fee premium and 1.42% higher fraud risk
- Top 3 high-fee merchant categories: Digital Goods (3.8%), Travel Services (3.2%), Electronics (2.9%)
- Top 3 high-fraud categories: Digital Goods (4.2%), Gift Cards (3.8%), Luxury Goods (2.7%)

#### 2. Fee Structure Analysis
**Key drivers of payment processing fees:**
- **Merchant Category:** Digital goods merchants pay highest fees (3.8% average)
- **Transaction Size:** Smaller transactions (<$50) incur highest relative fees (3.1% vs 1.6% for >$1000)
- **Card Type:** Premium cards (Platinum/Black) cost merchants 0.7-1.2% more than standard cards
- **Cross-Border Premium:** International transactions cost 0.83% more than domestic
- **Acquirer Location:** Transactions processed through certain countries (CY, MT, LU) show 0.5-0.7% higher fees

**Fee Optimization Opportunities:**
1. Implement category-specific pricing tiers
2. Bundle small transactions (<$50) into larger settlements
3. Negotiate lower rates for premium card transactions
4. Develop domestic processing alternatives for cross-border transactions

#### 3. Fraud Pattern Analysis
**High-risk factors:**
- **Digital goods** show 4.2% fraud rate (4.3× overall average)
- **Cross-border transactions** have 1.42% higher fraud incidence
- **Larger transactions** (>$500) show 2.1% fraud rate vs 0.8% for <$100
- **Card types:** Virtual/prepaid cards show 1.8× higher fraud rates
- **CVV absence:** Transactions without CVV have 3.2× higher fraud risk

**Fraud Reduction Strategies:**
1. Implement enhanced verification for digital goods merchants
2. Apply transaction limits and velocity checks for cross-border payments
3. Require CVV verification for transactions >$200
4. Develop ML-based fraud scoring models targeting high-risk categories

#### 4. Acquirer Selection Analysis
**Cost-Risk Tradeoffs by Location:**

\| Acquirer Country \| Avg Fee \| Fraud Rate \| Volume \|
\|---\|---\|---\|---\|
\| Germany (DE) \| 2.05% \| 0.82% \| 32% \|
\| France (FR) \| 2.12% \| 0.91% \| 28% \|
\| Cyprus (CY) \| 2.63% \| 1.24% \| 8% \|
\| Luxembourg (LU) \| 2.58% \| 1.31% \| 6% \|

**Acquirer Optimization Strategies:**
1. Shift volume from high-cost acquirers (CY/LU) to DE/FR where possible
2. Implement multi-acquirer routing based on transaction profile
3. Negotiate volume-based discounts with primary acquirers
4. Develop failover systems to maintain uptime during routing changes

#### 5. Merchant-Specific Recommendations

**Digital Goods Merchants (High Fee, High Fraud):**
- Implement 3D Secure for all transactions
- Negotiate flat-rate pricing instead of percentage-based
- Introduce micro-transaction pricing tiers
- Deploy behavioral biometrics for fraud prevention

**Retail Merchants (Medium Fee, Low Fraud):**
- Implement transaction size-based fee optimization
- Encourage standard card usage through loyalty programs
- Batch small transactions for processing efficiency
- Adopt address verification for high-value items

**Service Providers (Low Fee, Medium Fraud):**
- Implement recurring billing optimization
- Develop CVV-required policy for new customers
- Introduce cross-border fee discounts
- Use predictive analytics for chargeback prevention

#### 6. Implementation Roadmap
1. **Phase 1 (0-3 months):** Implement high-impact fee optimizations (acquirer routing, category-based pricing)
2. **Phase 2 (3-6 months):** Deploy fraud reduction systems (CVV enforcement, ML scoring)
3. **Phase 3 (6-12 months):** Develop advanced optimization (dynamic pricing, behavioral biometrics)

**Expected Results:**
- 18-22% reduction in payment processing costs ($4.2-5.1M annually)
- 25-30% reduction in fraud losses ($1.8-2.2M annually)
- 1.5-2% improvement in conversion rates through optimized checkout flows
- Enhanced merchant satisfaction through tailored payment solutions

This comprehensive analysis provides actionable pathways to optimize payment processing economics while maintaining security and conversion performance. Implementation should be prioritized based on merchant-specific profiles and transaction patterns. | Below is the comprehensive report generated by our analysis:

---
Data Science Report on Payment Transaction Dataset Analysis
---

Overview:
This analysis examined fee structures, fraud patterns, and transaction characteristics from the payment transaction dataset to identify strategies for cost optimization while preserving security and conversion rates. Multiple data sources—fee details (fees.json), acquirer information (acquirer_countries.csv), merchant profiles (merchant_data.json), merchant category codes (merchant_category_codes.csv), and transaction records (payments.csv)—were integrated for a robust view.

Key Findings:
---
1. Fee Structure Drivers:
- Summary statistics indicate that fees correlate strongly with transaction amounts. High-value transactions incur higher absolute fees, despite relatively stable fee percentages. This pattern suggests that the fee structure is largely volume-driven.
- Fees.json data reveal that some acquirers apply additional fixed charges, exposing opportunities for negotiations or alternative pricing models.

2. Fraud Pattern Analysis:
- Transactions flagged as fraudulent exhibit higher fee volatility and, occasionally, an elevated fee-to-amount ratio. This may stem from extra chargeback or remediation costs.
- By isolating transactions with fraud flags, merchants can explore risk-based pricing strategies that weigh fraud prevention costs against processing efficiency.

3. Acquirer Selection Strategies:
- Analysis of fee performance by acquirer shows significant variability. Certain acquirers, especially those active in lower-risk regions, offer more competitive fee structures.
- When acquirer data is merged with country-level information, geographical differences in fee formulations become apparent. This analysis suggests that merchants should strategically select acquirers based on their target market's risk profile and fee competitiveness.

4. Merchant Category and Profile Insights:
- Merging merchant transaction data with merchant category codes indicates that some merchant categories (e.g., high-ticket retail or service sectors) face disproportionately high fees.
- Merchant profile variables such as business size, transaction volume, and risk ratings (obtained from merchant_data.json) greatly influence fee structures and fraud vulnerability. Tailored strategies are thus needed depending on the merchant profile.

Actionable Recommendations:
---
1. Optimizing Fee Structures:
- Negotiate with acquirers to reduce fixed fees, particularly for high-volume transactions. Consider shifting to pricing models that reward transaction efficiency.
- Explore dynamic fee assessments where lower-risk, high-value transactions benefit from a reduced per-transaction fee.

2. Fraud Risk Mitigation:
- Enhance fraud detection methods, especially for segments identified as high risk, potentially using machine learning-based anomaly detection before transactions are processed.
- Implement risk-based fee adjustments that incorporate security costs for transactions with higher fraud potential.

3. Tailored Acquirer Selection:
- For merchants with international operations, perform data-driven assessments to select acquirers with optimal fee structures within target markets.
- Continuously monitor acquirer performance (fee levels, fraud management, dispute resolution) to ensure ongoing cost effectiveness.

4. Merchant-Specific Strategies:
- Small and medium-sized merchants should prioritize robust fraud prevention tools to reduce exposure while controlling costs.
- High-volume merchants can leverage their scale for negotiating volume discounts or dedicated agreements that improve fee structures.

Conclusion:
---
The analysis indicates that careful management of fee structures and acquirer selection—combined with proactive fraud detection—can result in significant cost savings without compromising security or conversion rates. Tailoring these recommendations based on merchant-specific and regional nuances offers practical pathways for reducing overall payment processing costs.

Next Steps:
---
- Further investigate temporal trends to refine dynamic pricing strategies.
- Invest in machine learning models for real-time fraud prediction and transaction scoring.
- Regularly re-assess acquirer agreements and fee strategies in light of evolving transaction data.

By implementing these recommendations, merchants can optimize payment processing fees while maintaining robust security and enhancing conversion outcomes. |

*Figure 12.* A data analysis case of autonomous data science, from data sources to research reports.

| Instruction | DeepAnalyze-8B | Deepseek-r1 | o3-mini |
|---|---|---|---|
| Analyze the provided payments dataset and supporting documentation to produce a comprehensive data insight report. The report should identify the primary drivers of transaction costs and fraud risk for the merchants. Investigate how factors such as transaction characteristics (e.g., ACI, card scheme, amount), merchant profiles (e.g., account type, MCC, acquirer choice), and behavioral patterns (e.g., time of day, cross-border activity) influence processing fees and fraudulent disputes. The goal is to derive actionable insights that can help merchants optimize their payment configurations and operational practices to minimize costs and mitigate risk. | # Comprehensive Analysis of Payment Processing Fees and Fraud Risk Factors

## Executive Summary

This comprehensive analysis examines the intricate relationship between payment transaction characteristics, merchant profiles, and resulting processing fees and fraud risk. Through systematic analysis of over 23 million payment transactions across 30 merchant accounts, we identify key drivers of transaction costs and fraud risk. The study reveals that card scheme selection accounts for 68% of fee variation, while merchant category codes influence 42% of fraud risk profiles. Our findings demonstrate that strategic configuration of payment processing parameters can reduce costs by 18-22% while simultaneously decreasing fraudulent dispute rates by 14-17%.

## Methodology and Data Overview

The analysis leverages a comprehensive dataset comprising 23,028,700 payment transactions processed through multiple acquirer networks between January 2022 and December 2023. The dataset includes detailed transaction-level attributes including card scheme (Visa, Mastercard, Amex, etc.), authorization characteristics (ACI codes), merchant category codes, and transaction timing patterns. Merchant profiles include account types (Platform - Franchise, Retail, etc.), capture delay settings, and historical fraud patterns. Fee structures were extracted from 1,000 distinct fee rules governing processing costs.

Statistical analysis employed multivariate regression modeling, decision tree analysis for fraud pattern identification, and cost-benefit optimization modeling. Machine learning techniques including random forest classification achieved 89.3% accuracy in predicting fraudulent transactions when incorporating both transaction attributes and merchant profiles. The analysis controlled for seasonal variations, economic conditions, and geographic factors affecting payment patterns.

## Key Findings on Cost Drivers

### Card Scheme and Authorization Characteristics

Card scheme selection emerges as the most significant determinant of processing costs, accounting for 68% of fee variation according to our regression analysis. Visa transactions averaged €0.1154 in fees compared to €0.0833 for debit card transactions (p<0.001). Authorization Characteristics Indicators (ACI) showed particularly strong cost implications, with ACI 'B' transactions (Card Not Present - 3-D Secure) incurring 23% higher fees than ACI 'A' (Card Present) transactions. The table below illustrates the cost differential by authorization type:

\| ACI Code \| Description \| Fixed Fee (€) \| Variable Rate (bps) \|
\|--------- \| ------------ \| --------------- \| -------------------- \|
\| A \| Card Present \| 0.092 \| 38.2 \|
\| B \| 3-D Secure \| 0.115 \| 42.7 \|
\| C \| Tokenized \| 0.103 \| 40.1 \|
\| D \| Card On File \| 0.121 \| 45.3 \|
\| E \| Recurring \| 0.087 \| 36.8 \|

Merchant category codes demonstrated significant impact on both costs and fraud risk. Retail merchants (MCC 5812) experienced 28% higher fraud rates than hospitality merchants (MCC 7999), while wholesale distributors (MCC 7372) enjoyed 19% lower dispute rates. Account type proved equally impactful, with Platform Franchise accounts incurring 12% lower fees than Retail accounts for equivalent transactions. The analysis identified three optimal fee optimization opportunities yielding 18-22% cost savings through strategic configuration of payment parameters.

## Fraud Risk Patterns and Mitigation Strategies

Fraud risk patterns revealed distinct behavioral signatures across merchant segments. Cross-border transactions (>15% of volume from non-primary markets) showed 37% higher fraud rates, while transactions occurring outside normal business hours demonstrated 29% elevated risk. Machine learning models identified four key fraud indicators with predictive accuracy exceeding 85%:

1. Device fingerprint mismatch (92% predictive power)
2. IP country mismatch (87% predictive power)
3. Transaction velocity anomalies (83% predictive power)
4. Historical fraud profile deviation (81% predictive power)

Merchant-specific risk profiles varied significantly by account type and category. Platform Franchise accounts demonstrated 19% lower fraud rates than Retail accounts for equivalent transaction profiles, while Hospitality merchants showed 23% better fraud prevention outcomes than Retail merchants. The analysis identified three optimal fraud prevention strategies yielding 14-17% reduction in fraudulent dispute rates through behavioral pattern optimization.

## Strategic Recommendations

### Cost Optimization Opportunities

Strategic configuration of payment parameters can yield significant cost savings while maintaining service quality. Our analysis identifies three primary optimization opportunities:

1. Authorization Strategy Optimization:
- Shift 30% of ACI 'D' (Card On File) transactions to ACI 'E' (Recurring) for 12-15% cost reduction
- Implement tokenization for 60% of card-not-present transactions to reduce fees by 8-10%
- Consolidate card schemes to two optimal options per merchant profile

2. Merchant Configuration Optimization:
- Align capture delay settings with business cycles (18% cost reduction)
- Optimize acquirer bank selection based on volume tiers (12-15% savings)
- Implement volume-based pricing tiers (9-11% savings)

3. Fraud Prevention Optimization:
- Implement behavioral pattern monitoring (14-17% fraud reduction)
- Optimize fraud rules by merchant segment (12-15% reduction)
- Implement dynamic fraud thresholds (10-13% reduction)

The comprehensive cost-benefit analysis demonstrates that implementation of these strategies can yield 22-25% reduction in processing costs while simultaneously reducing fraudulent dispute rates by 17-20%. The analysis further identifies specific configuration recommendations for each merchant segment based on account type, volume profile, and risk tolerance.

## Conclusion and Future Research Directions

This comprehensive analysis demonstrates the significant impact of strategic payment processing configuration on both cost efficiency and risk management. The findings challenge conventional wisdom regarding fixed fee structures, demonstrating that dynamic, data-driven configuration of payment parameters can yield substantial improvements in both financial and risk outcomes. Future research should explore the integration of emerging technologies including AI-driven authorization decisioning and real-time fraud prevention systems. The demonstrated methodologies provide a framework for continuous optimization of payment processing strategies in an increasingly complex and competitive marketplace. | Based on the analysis performed, here are the key insights into transaction cost drivers and fraud risk factors:

### 1. Primary Transaction Cost Drivers
- **Card Scheme Impact**: Visa transactions show significantly higher average fees (€X.XX) compared to Mastercard (€X.XX)
- **Cross-Border Premium**: Cross-border transactions incur XX% higher fees on average compared to domestic transactions
- **Account Type Differences**: Premium merchant accounts show XX% lower fee percentages than standard accounts
- **ACI Influence**: High ACI scores correlate with XX% lower fees due to reduced risk surcharges

### 2. Fraud Risk Patterns
- **Temporal Patterns**: Highest dispute rates occur between [XX:XX-XX:XX] local time
- **Card Brand Variation**: [Card Brand] shows XX% higher dispute rates than others
- **MCC Risk Profile**: MCC codes [X] and [Y] show XX% higher fraud rates than average
- **Cross-Border Risk**: Cross-border transactions have XX% higher dispute rates

### 3. Fee Optimization Opportunities
- **MCC-Scheme Combinations**: [Scheme] + [MCC Description] shows XX% lower fees than alternatives
- **Acquirer Selection**: Transactions routed through [Acquirer] show XX% lower fees
- **Volume Discounts**: Merchants processing >€XX/month qualify for XX% fee reductions

### 4. Operational Recommendations
1. Implement dynamic routing to prioritize [Scheme] for MCC [X] transactions
2. Add fraud screening enhancements during high-risk hours [XX:XX-XX:XX]
3. Migrate high-volume merchants to premium accounts for fee savings
4. Implement geolocation checks to flag high-risk cross-border patterns

The analysis reveals significant optimization potential through strategic payment configuration adjustments and targeted fraud controls. Further segmentation by merchant size and transaction volume could yield additional insights. | Below is a summary of the insights derived from our exploratory data analysis of the payments dataset and supporting documentation:

1. Transaction Characteristics & Card Scheme

• Our aggregation by card scheme shows that different brands can exhibit distinct behaviors. For instance, some card schemes are associated with higher average transaction amounts while others show a higher incidence (fraud rate) of fraudulent disputes. This suggests that merchants accepting cards from schemes with higher fraud rates may face increased processing fees due to the additional risk.

• Recommendation: Merchants can investigate alternative card acceptance strategies (or enforce stricter verification methods) for cards linked to historically higher fraud incidences; additionally, pricing or insurance adjustments may be considered.

2. Behavioral Patterns: Time of Day

• Analysis by the hour of the day indicates that fraud incidents are not uniformly distributed. Certain time slots (hours with higher fraud rates) are linked either to lower volumes or less predictability in transactions. Such time windows could be targeted for enhanced monitoring and fraud prevention controls.

• Recommendation: Merchant operations can consider operating additional fraud screening measures during peak fraud risk hours, optimizing staffing decisions or real-time risk analytics at these times.

3. Cross-Border Activity

• When investigating the role of cross-border versus domestic transactions, the data reveals that cross-border transactions tend to have different fee profiles and may encounter higher fraud rates. This may be due to additional banking and regulatory complexities or differences in consumer verification processes across countries.

• Recommendation: Merchants with substantial cross-border activity should focus on localizing fraud prevention mechanisms, possibly integrating additional identity verification tools tailored to international transactions, and reassessing fee configurations offered by acquirers regarding international payments.

4. Merchant Profiles and Operational Practices

• By merging merchant profiles (from merchant_data.json) with transactions data, we identified that merchant account type influences both transaction volumes and fraud incidence. For instance, certain account types (potentially tied to higher risk industry classifications or MCC codes) exhibit higher fraud rates alongside varying average transaction amounts.

• Recommendation: Merchants should review their merchant account profiles and consider if their current configurations, acquirer choices, and business practices (such as transaction volume patterns) are aligned with their risk tolerance. Further collaboration with acquirers on fee structures that account for these risk profiles might help optimize processing costs.

5. Additional Considerations: Fee Structures

• Although the direct linkage between fee configuration (from fees.json) and transaction characteristics was not explicitly modeled due to integration challenges, the available fee data suggests that fee structures can vary based on parameters such as card scheme, transaction amount, and risk profile.

• Recommendation: A further detailed analysis that integrates the fee configurations with live transactional data can uncover more granular relationships. Merchants may leverage such insights to negotiate better fee agreements or adjust internal practices (e.g., reducing cross-border fraud through better verification) that, in turn, optimize their fee liabilities.

Final Summary:

The analysis indicates that transaction characteristics (e.g. card scheme and transaction amount), behavioral patterns (e.g. time-of-day and cross-border activity), and merchant-specific factors (account type, MCC) all contribute as primary drivers of processing fees and fraud risk. Merchants are advised to:
• Review card scheme mixtures and target risk mitigation strategies for higher-risk cards.
• Enhance fraud monitoring during identified high-risk hours.
• Implement tighter fraud controls for cross-border transactions.
• Reassess merchant profiles and operational configurations to align with fee structures and risk profiles.
These insights offer actionable directions for merchants aiming to optimize their payment configurations while controlling costs and mitigating risks.

This comprehensive view reinforces that an integrated approach, using both transaction and merchant profile data alongside fee configurations, is critical for reducing exposure to fraud and managing processing costs effectively. |

*Figure 13.* A data insight case of autonomous data science, from data sources to research reports.

| Instruction | DeepAnalyze-8B | Deepseek-r1 | o3-mini |
|---|---|---|---|
| Based on the provided data (manual.md, payments.csv, merchant_data.json, fees.json, acquirer_countries.csv, merchant_category_codes.csv), generate a comprehensive **Data Report** analyzing the payment processing operations for the merchants. The report must follow the exact structure and content outline specified below.

# Strategic Recommendations Report: Payment Processing Analysis

## 1. Executive Summary
- Provide a high-level overview of key findings and most critical recommendations.
- Summarize overall performance, cost efficiency, and fraud risk levels.

## 2. Introduction
- State the purpose and scope of the analysis.
- Briefly describe the merchants and time period covered.

## 3. Data Sources and Quality
- List and describe all data sources used in the analysis.
- Assess data completeness, accuracy, and any limitations encountered.

## 4. Analytical Approach
- Explain the methodology used for fee calculation and cost analysis.
- Describe how fraud risk and processing efficiency were evaluated.
- Outline the criteria for identifying optimization opportunities.

## 5. Findings
### 5.1 Transaction Volume and Patterns
- Analyze transaction volumes by merchant, time, and card scheme.
- Identify seasonal trends and peak processing periods.

### 5.2 Fee Structure Analysis
- Calculate and compare processing costs across different merchants.
- Identify the most expensive fee components and their drivers.
- Analyze how merchant characteristics affect fee rates.

### 5.3 Fraud and Risk Assessment
- Evaluate fraud rates and dispute patterns.
- Assess the effectiveness of current fraud prevention measures.
- Identify high-risk transaction types and merchants.

### 5.4 Processing Efficiency
- Analyze authorization rates and refusal patterns.
- Evaluate acquirer selection and routing efficiency.
- Assess capture delay practices and their cost implications.

## 6. Limitations
- Discuss any data constraints or methodological limitations.
- Identify areas where additional data would improve analysis.

## 7. Strategic Recommendations
### 7.1 Cost Optimization Strategies
- Specific recommendations for reducing processing fees.
- Suggestions for optimizing transaction routing and acquirer selection.

### 7.2 Fraud Prevention Improvements
- Recommendations for enhancing security measures.
- Strategies for reducing fraud-related costs and chargebacks.

### 7.3 Operational Efficiency Enhancements
- Suggestions for improving authorization rates and reducing refusals.
- Recommendations for optimizing capture timing and settlement processes.

## 8. Conclusion
- Summarize key insights and expected benefits of implemented recommendations.
- Outline next steps for implementation and monitoring.

## 9. Bibliography
- List all referenced documents, manuals, and data sources.

## 10. Appendix
- Include supporting tables, charts, and detailed calculations.
- Provide merchant-specific analysis and recommendations.

Ensure the report is data-driven, with specific metrics and calculations supporting all findings and recommendations. Use clear, professional language suitable for business stakeholders. | # Strategic Recommendations Report: Payment Processing Analysis

## 1. Executive Summary
This comprehensive analysis of payment processing operations reveals significant opportunities for cost optimization and operational improvement. The dataset encompasses 1,000 transactions across 30 merchants, with an average processing fee of €0.6574 per transaction, comprising €0.8058 in fixed fees and €6.3354 in variable rate fees per €10,000 processed. The analysis identifies three primary areas for strategic improvement: fee structure optimization (35% potential savings), fraud prevention enhancement (22% reduction target), and operational efficiency gains (15% improvement). Merchant segmentation reveals significant variation in fee structures, with cross-border transactions averaging 18% higher fees than domestic transactions. The data demonstrates that 23% of transactions exhibit characteristics warranting enhanced fraud controls, while 17% of merchants operate with suboptimal acquirer relationships. These findings enable targeted interventions to improve profitability while maintaining service quality.

## 2. Introduction
This report analyzes payment processing operations for a network of 30 merchants across multiple European countries, covering transaction data from 2023. The analysis incorporates transactional records, merchant attributes, fee structures, and acquirer relationships to evaluate operational efficiency and identify optimization opportunities. The dataset includes 1,000 sampled transactions with an average value of €87.42, representing €87,420 in total processed volume. Merchant categories span retail (42%), hospitality (23%), and professional services (35%), with merchant category codes following ISO 8583 standards. The analysis period covers a complete calendar year, capturing seasonal patterns and business cycle effects. Data quality assessment indicates 98.7% completeness in critical fields, with missing values concentrated in optional fields like email addresses (1.3% missing). The methodology combines quantitative analysis of transaction patterns with rule-based fee calculations against the 1,000 fee rules in the system.

## 3. Data Sources and Quality
The analysis integrates six primary data sources, each contributing unique dimensions to the payment processing analysis. The payments dataset forms the transactional foundation, containing 21 fields capturing merchant, card scheme, timing, and amount details. Merchant attributes from merchant_data.json provide 5 key attributes per merchant, while fees.json contains 1,000 complex fee rules with 12 conditional fields each. Merchant category codes provide standardized industry classifications, and acquirer countries map 8 payment processors to their respective countries. Data quality assessment reveals excellent completeness (98.2% complete records) with minimal missing values concentrated in optional fields. The fee rules demonstrate comprehensive coverage, with 92% of transactions matching at least one fee rule and 78% matching exactly one rule. Validation against business rules in manual.md confirms structural compliance with payment processing standards. Limitations include the 1,000-transaction sample size and lack of historical comparisons, though the analysis accounts for these through statistical techniques.

## 4. Analytical Approach
The methodology combines descriptive analytics, predictive modeling, and prescriptive optimization to evaluate payment processing operations. Fee calculations implement the formula: Fee = FixedAmount + (Rate × TransactionValue / 10,000), applying the fee matching rule from 12 conditional dimensions. Fraud risk assessment combines transaction velocity, geographic patterns, and historical fraud rates using a logistic regression model with AUC-ROC of 0.82. Processing efficiency metrics include authorization rates (87.3%), refusal rates (9.1%), and capture delay patterns. Optimization models employ linear programming to identify cost-saving improvements, achieving 14% simulated savings. Merchant segmentation uses k-means clustering (k=5) on transaction patterns and fee structures. All analyses implement 5-fold cross-validation to ensure robustness, with key metrics showing <5% variation across folds. The approach balances statistical rigor with business interpretability, ensuring actionable insights.

## 5. Findings
### 5.1 Transaction Volume and Patterns
Transaction analysis reveals distinct patterns across merchants, card schemes, and time periods. Volume peaks between 11am-2pm (28% of daily volume) and 4-9pm (31%), with weekday volumes 42% higher than weekends. Cross-border transactions (issuing country ≠ acquirer country) represent 38% of volume but 53% of fees, averaging €0.72/transaction versus €0.61 for domestic. Card scheme analysis shows GlobalCard processing 22% more transactions than competitors, though with 18% higher fees. Merchant category analysis reveals retail merchants process 3.2 transactions/merchant versus 1.8 for professional services, with average transaction values of €78.42 and €124.57 respectively. Fraud patterns show 2.1% overall fraud rate, but 4.7% for high-value (>€500) credit card transactions. Seasonal analysis identifies November peaks (18% above average) and February troughs (23% below average).

### 5.2 Fee Structure Analysis
Fee analysis reveals significant variation across merchants and conditions. The average fee structure contains 4.2 conditional dimensions per rule, with capture delay (32% of rules), merchant category (28%), and card scheme (22%) being most common conditions. Fixed fees range from €0.05 to €0.25 (mean €0.12), while variable rates range from 15 to 95 basis points (mean 42 bps). Merchant segmentation identifies three distinct fee profiles: low-volume/high-margin (12 merchants), volume-driven (15 merchants), and cross-border focused (3 merchants). Fee optimization modeling identifies potential savings of €1,200 annually through acquirer routing changes alone. Intracountry transactions enjoy 18% lower fees on average, suggesting substantial routing optimization opportunities. The analysis identifies 7 merchants operating with suboptimal acquirer relationships, representing €8,400 in potential savings.

### 5.3 Fraud and Risk Assessment
Fraud analysis identifies four high-risk merchant categories: travel (7.2% fraud rate), electronics (6.8%), retail (5.1%), and hospitality (4.9%). Transaction velocity analysis flags 23% of transactions exceeding historical norms, while geographic analysis identifies 17% of transactions with unusual country mismatches. Machine learning models predict fraud with 82% accuracy, identifying 14% of transactions requiring manual review. Capture delay analysis shows merchants with >7 day delays experience 38% higher chargebacks. Fraud prevention improvements could reduce losses by €1,200 annually through enhanced velocity checks and 3D Secure implementation. The analysis identifies 8 merchants with suboptimal fraud controls, representing €9,800 in potential savings.

### 5.4 Processing Efficiency
Authorization analysis reveals 87.3% authorization rates, with debit cards (92%) outperforming credit (82%). Refusal analysis identifies acquirer routing as a primary cause (63%), followed by card scheme restrictions (22%) and merchant limits (15%). Capture delay analysis shows merchants with <3 day delays achieve 28% lower dispute rates. Settlement timing analysis identifies 17% of merchants settling >T+5, correlating with 31% higher late fees. Routing optimization modeling identifies €1,400 in potential savings through acquirer reassignment. Authorization rate improvements could yield €2,100 annually through improved merchant training. The analysis identifies 12 merchants with suboptimal processing practices, representing €15,200 in potential savings.

## 6. Limitations
The analysis faces several constraints requiring cautious interpretation. The 1,000-transaction sample, while representative, may not capture extreme values adequately. The static fee rules don't account for dynamic pricing, potentially underestimating elasticity. Fraud models lack real-time transaction data, potentially missing emerging patterns. The merchant segmentation uses only 5 features, potentially oversimplifying complex business models. The analysis period covers one calendar year, limiting seasonal comparisons. Data limitations include missing merchant bank account details and limited historical benchmarking. The optimization models assume static merchant volumes, while actual volumes may fluctuate. These limitations are mitigated through conservative confidence intervals and sensitivity analysis. Future research could incorporate real-time data streams and machine learning for dynamic pricing.

## 7. Strategic Recommendations
### 7.1 Cost Optimization Strategies
Fee optimization should prioritize three initiatives: acquirer routing (€1,400 potential), fee structure review (€1,200), and capture delay reduction (€1,500). Merchant-specific analysis identifies 7 merchants with €8,400 in potential savings with €9,800 in fraud prevention improvements. The analysis recommends implementing dynamic routing algorithms to capture 12% of potential savings automatically. Fee structure review should focus on cross-border transactions, where fees average 18% higher. Capture delay improvements should target merchants with >7 day delays, potentially saving €2,100 annually. The implementation roadmap prioritizes quick wins (routing) before complex changes (fee restructuring). The projected 28% annual savings of €38,900 delivers 3:1 ROI within 6 months. The program should include ongoing monitoring with monthly KPIs and quarterly reviews.

### 7.2 Fraud Prevention Improvements
Fraud prevention should prioritize three areas: velocity controls (€1,200 potential), 3D Secure implementation (€900), and merchant training (€700). The analysis identifies 8 merchants with €9,800 in fraud prevention improvements. Velocity controls should focus on high-value transactions (>€500 transactions), where fraud rates peak at 4.7%. 3D Secure implementation should target credit card transactions, where fraud rates peak at 5.1%. Merchant training should focus on high-risk categories (travel, electronics). The implementation roadmap prioritizes quick wins (velocity controls) before complex changes (3D Secure). The projected 22% reduction in fraud losses (€2,100) delivers 3:1 ROI within 9 months. The program should include real-time monitoring with daily dashboards and weekly reviews.

### 7.3 Operational Efficiency Enhancements
Processing efficiency should prioritize three areas: authorization rates (€2,100 potential), settlement timing (€1,500), and acquirer selection (€1,400). The analysis identifies 12 merchants with €15,200 in processing improvements. Authorization rate improvements should focus on credit card transactions, where rates lag debit by 10 percentage points. Settlement timing should target merchants with >T+5 delays, where fees peak at 31% higher. Acquirer selection should focus on cross-border transactions, where fees peak at 18% higher. The implementation roadmap prioritizes quick wins (authorization training) before complex changes (acquirer reassignment). The projected 15% improvement (€22,700) delivers 3:1 ROI within 12 months. The program should include real-time monitoring with daily dashboards and monthly reviews.

## 8. Conclusion
This comprehensive analysis identifies substantial opportunities to improve payment processing operations through data-driven optimization. The findings demonstrate that strategic interventions can yield €38,900 in annual savings (28% of current costs) while improving service quality. The recommended initiatives combine quick wins with longer-term transformations, delivering measurable improvements across cost, risk, and efficiency metrics. Implementation requires cross-functional collaboration between operations, risk, and finance teams. The proposed roadmap balances immediate action with sustainable transformation, ensuring lasting impact. The analysis provides the foundation for ongoing performance management, with recommended metrics for continuous improvement. Future research should incorporate real-time data streams and machine learning for dynamic optimization. The findings demonstrate the transformative potential of data analytics in payment processing operations.

## 9. Bibliography
Merchant Guide to Optimizing Payment Processing and Minimizing Fees (v2.1, November 2024)
Payment Processing Standards (ISO 8583:2021)
Merchant Category Code Reference (v3.2, October 2024)
Acquirer Country Mapping (v1.5, September 2024)
Fee Rule Specifications (v2.0, August 2024)
Data Dictionary (v1.0, July 2024)

## 10. Appendix
Appendix A: Merchant Segmentation Details
Appendix B: Fee Rule Matching Algorithm
Appendix C: Fraud Model Specifications
Appendix D: Optimization Model Parameters
Appendix E: Implementation Roadmaps
Appendix F: Detailed Merchant Analysis
Appendix G: Technical Specifications | I'll analyze the data and prepare the comprehensive report based on the code execution results. Here's the report:

### Strategic Recommendations Report: Payment Processing Analysis

#### 1. Executive Summary
- Overall transaction processing shows moderate efficiency with an authorization rate of 82.3%
- Fraud rate is concerning at 3.5%, significantly impacting revenue
- Refusal rate of 1.2% indicates potential customer satisfaction issues
- Highest costs come from interchange fees (average 0.45% per transaction)
- Capture delay shows room for improvement with average 1.8 hours between creation and capture

#### 2. Introduction
- Analysis covers payment processing for multiple merchants over an unspecified period
- Purpose: Identify cost optimization, fraud reduction, and efficiency improvements
- Scope: Transaction patterns, fee structures, fraud assessment, and processing efficiency

#### 3. Data Sources and Quality
- **payments.csv (23MB):** Core transaction data with authorization status, fraud flags, and timestamps
- **merchant_data.json (6.7KB):** Merchant profiles including IDs and categories
- **fees.json (518.7KB):** Detailed fee structures by merchant and transaction type
- **acquirer_countries.csv (0.2KB):** Acquirer country codes and names
- **merchant_category_codes.csv (26KB):** MCC category descriptions
- **Limitations:** Date range not specified in sample data; merchant name not included

#### 4. Analytical Approach
- Fee calculation: Summed interchange, scheme, and acquirer fee components
- Fraud risk: Calculated fraud and dispute rates; identified high-risk merchants
- Processing efficiency: Analyzed authorization rates, refusal reasons, and capture delays
- Optimization criteria: Cost reduction potential, fraud mitigation impact, and efficiency gains

#### 5. Findings
##### 5.1 Transaction Volume and Patterns
- Transaction distribution:
  - Visa: 38% of transactions
  - Mastercard: 35%
  - American Express: 27%
- Strong seasonal patterns: 15% higher volume in December
- Daily peak: 3-4 PM (18% of daily transactions)

##### 5.2 Fee Structure Analysis
- Most expensive components:
  1. Cross-border interchange fee (0.32%)
  2. Scheme fee (0.25%)
  3. Acquirer markup (0.18%)
- Fee variations by merchant size: Smaller merchants pay 22% higher fees on average
- High-MCC merchants (hotels, car rentals) incur 30% higher fees

##### 5.3 Fraud and Risk Assessment
- Highest fraud rates by merchant:
  - Merchant X: 8.2%
  - Merchant Y: 6.7%
  - Merchant Z: 5.9%
- Card scheme fraud rates:
  - American Express: 4.8%
  - Visa: 3.1%
  - Mastercard: 2.9%

##### 5.4 Processing Efficiency
- Authorization rate: 82.3%
- Top refusal reasons:
  - Insufficient funds: 45%
  - Suspected fraud: 30%
  - Invalid card: 15%
- Capture delay statistics:
  - Mean: 1.8 hours
  - Max: 48 hours
  - 90th percentile: 5 hours

##### 6. Limitations
- No transaction amount data for precise revenue impact calculation
- Limited merchant profiles for category-level analysis
- Time period not specified limits trend analysis
- Missing chargeback data for complete fraud assessment

##### 7. Strategic Recommendations
###### 7.1 Cost Optimization Strategies
- Negotiate interchange fees for high-volume merchants (potential 15% savings)
- Implement MCC optimization for high-cost categories
- Consolidate acquirer relationships to reduce markup fees

###### 7.2 Fraud Prevention Improvements
- Introduce machine learning fraud scoring for high-risk merchants
- Implement 3D Secure for American Express transactions
- Create merchant-specific fraud thresholds based on historical patterns

###### 7.3 Operational Efficiency Enhancements
- Optimize transaction routing to acquirers with >90% authorization rates
- Implement automated retry system for \"insufficient funds\" declines
- Reduce capture delay target to 1 hour (potential 0.2% fee reduction)

###### 8. Conclusion
- Implementing recommendations could reduce processing costs by 18-22%
- Fraud reduction initiatives could save $500K+ annually for mid-sized merchants
- Efficiency improvements could increase revenue capture by 5-7%
- Next steps: Prioritize cost optimization and fraud prevention for highest-impact merchants

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

###### 10. Appendix
- Detailed fee calculation examples available upon request
- Fraud pattern charts by card scheme and time of day
- Capture delay distribution histogram
- Acquirer performance comparison table

This report provides data-driven insights to optimize payment operations, reduce costs, and mitigate fraud risks across your merchant portfolio. | Below is the comprehensive Data Report generated based on the analysis of the provided payment processing data:

Strategic Recommendations Report: Payment Processing Analysis

1. Executive Summary
─────────────────────
- Key Findings:
  - Total Transactions Processed: [Extracted from payments.csv]
  - Merchants Covered: [Extracted from payments.csv]
  - Significant variation noted in fee components and processing times.
- Recommendations:
  - Optimize acquirer selection and fee structures.
  - Enhance fraud detection measures for high-risk transaction types.

2. Introduction
─────────────────────
- Purpose: To analyze payment processing operations with an emphasis on cost efficiency, risk, and operational performance.
- Scope: The analysis covers multiple merchants over the time period as indicated in the transaction data.

3. Data Sources and Quality
─────────────────────
- Data Sources:
  - manual.md: Operational procedures and system guidelines.
  - payments.csv: Transaction records with detailed payment information.
  - merchant_data.json: Profiles and characteristics of merchants.
  - fees.json: Detailed fee structures per transaction and merchant.
  - acquirer_countries.csv: Geographical distribution of acquirers.
  - merchant_category_codes.csv: Classification of merchants by business types.
- Data Quality Assessment:
  - Payments data is extensive but has some missing fields (e.g., transaction dates) that might impact temporal analyses.
  - Fees data provides detailed breakdowns; however, its complexity requires careful reconciliation with merchant profiles.
  - Overall, data is adequate for in-depth analysis, though supplementary granular data (especially on fraud incidents) would further enhance insights.

4. Analytical Approach
─────────────────────
- Methodology:
  - Fee Calculation: Aggregated fees per merchant were compared to reveal variations in cost efficiency. Major fee drivers included processing fees, acquirer service charges, and additional service fees.
  - Fraud Risk Evaluation: Analyzed fraud indicators via dispute records in the payments data correlated against merchant risk profiles from merchant_data.json.
  - Processing Efficiency: Evaluated key metrics like authorization and capture rates, refusal patterns, and acquirer routing efficiency using the payments.csv data.
  - Optimization Criteria: Identified high fee components, elevated fraud signals, and sub-optimal processing speeds as primary opportunities for operational improvements.

5. Findings
─────────────────────
5.1 Transaction Volume and Patterns
- Total transactions processed: Data from payments.csv shows a large volume across the analyzed period.
- Seasonal trends: Evidence of peaks during promotional periods and seasonal events.
- Card scheme insights: Diverse usage across major card networks (Visa, MasterCard, etc.) indicate varied consumer behaviors.

5.2 Fee Structure Analysis
- Variability in fee structures: Fees vary notably across merchants, influenced by operational volume and merchant category.
  - Major fee components: Processing fees and acquirer service charges represent the highest cost elements.
  - Impact of merchant characteristics: Analysis suggests that merchant category (as indexed in merchant_category_codes.csv) and volume are key determinants driving fee rates.

5.3 Fraud and Risk Assessment
- Fraud trends: Some segments, particularly those with high transaction volumes, show elevated dispute rates.
- Effectiveness of fraud prevention: Current measures are moderately effective; however, enhanced real-time monitoring could reduce fraud-related costs.
- High-risk identification: Certain transaction types and specific merchants are flagged for heightened fraud risk.

5.4 Processing Efficiency
- Authorization vs. refusal: Most merchants experience industry-standard authorization rates, though a subset shows higher refusal numbers.
  - Acquirer selection: Analysis based on acquirer_countries.csv reveals potential routing improvements that could lower fees.
- Capture delays: Delay in capture processes directly adds to cost overheads, suggesting a need for more timely settlement processes.

6. Limitations
─────────────────────
- Data Constraints: Some fields in payments.csv (such as transaction dates) were incomplete or inconsistently formatted, affecting the precision of time-based analyses.
- Methodological Limitations: Assumptions made during aggregation might not fully capture all fee nuances.
- Additional Data: Detailed data on fraud incidents and more granular merchant performance metrics would provide enhanced clarity.

7. Strategic Recommendations
─────────────────────
7.1 Cost Optimization Strategies
- Negotiate fee rates by leveraging merchant's high-volume statistics to secure better terms.
- Re-evaluate acquirer and transaction routing decisions to identify efficient cost reductions based on regional performance data from acquirer_countries.csv.

7.2 Fraud Prevention Improvements
- Enhance real-time fraud detection systems and update risk models using detailed merchant data.
- Implement targeted monitoring for high-risk transaction segments identified in the analysis.

7.3 Operational Efficiency Enhancements
- Streamline authorization processes by identifying and mitigating common refusal reasons.
- Reduce capture delays through improved system integration and proactive settlement workflows.

8. Conclusion
─────────────────────
- Key Insights: The analysis underscores the need for improvements in reducing processing fees, strengthening fraud prevention, and enhancing overall operational efficiency.
- Next Steps: Implement the recommended optimizations, closely monitor the outcomes with real-time dashboards, and continually refine fraud risk models.

9. Bibliography
─────────────────────
- manual.md
- payments-readme.md
- payments.csv
- merchant_data.json
- fees.json
- acquirer_countries.csv
- merchant_category_codes.csv

10. Appendix
─────────────────────
- Detailed tables and visualizations (e.g., fee breakdowns, seasonal transaction trends, and risk assessments) underpin the findings.
- Merchant-specific analysis and customized recommendations are available upon request.

This report is data-driven and built upon specific metrics and calculations derived from the provided data sources. The recommendations are aimed at reducing costs, mitigating fraud risk, and optimizing operational efficiency in the payment processing environment. |

*Figure 14.* A case of autonomous data science with report constraints.

| Instruction | DeepAnalyze-8B | Deepseek-r1 | o3-mini |
|---|---|---|---|
| Generate a report summarizing any notable trends, findings, or summaries in the data. | # Comprehensive Analysis of Payment Processing Data: Trends, Patterns, and Business Insights

## Executive Summary
This comprehensive analysis examines a rich dataset encompassing merchant payment transactions, fee structures, and merchant characteristics. The study reveals significant insights into payment processing patterns, fraud risks, and fee optimization opportunities. Through multidimensional analysis of over 138,000 transactions across 30 merchants, we identify actionable insights for payment processors and merchants alike. The findings demonstrate substantial variation in fraud rates (ranging from 1.2% to 18.7% across merchant categories), fee structures (varying from €0.01 to €0.14 per transaction), and transaction patterns (peak volumes at 12:00-13:00 hours).

## Merchant Category Analysis and Fraud Patterns

The merchant category analysis reveals significant variation in transaction volumes and fraud rates across different business sectors. The top merchant category (code 7997) accounts for 28.7% of transactions in our sample, followed by category 7993 (14.3%) and 7372 (14.1%). Fraud rates show substantial variation across categories, with the highest fraud rates observed in categories 5812 (18.7%) and 5942 (16.9%), compared to the overall average of 6.3%. These findings suggest that certain merchant categories require enhanced fraud prevention measures and potentially revised fee structures to account for higher risk profiles.

The temporal analysis of transactions reveals distinct patterns in payment processing. Transaction volumes peak at midday, with 28.4% of daily transactions occurring between 12:00-13:00 hours, suggesting that payment processing systems may benefit from capacity planning around these peak periods. The average transaction amount stands at €91.85, with a right-skewed distribution (median €55.74), indicating that while most transactions are relatively small, a significant minority involve substantial amounts exceeding €500.

## Fee Structure and Fee Optimization Opportunities

The fee structure analysis reveals complex, multi-dimensional pricing models that incorporate merchant category, transaction type, and risk factors. The average fee component consists of a fixed amount of €0.069 (SD €0.043) plus a variable rate of 54 basis points (SD 25.2 basis points). However, these averages mask significant variation across merchant categories, with certain high-risk categories facing fees up to 89 basis points. The data suggests that approximately 32% of fee rules apply universally (applicable to all merchant categories), while the remaining 68% show specific category targeting.

The relationship between fraud rates and fee structures reveals opportunities for optimized pricing. Merchants in high-fraud categories (fraud rates >15%) currently face an average fee premium of 18.7 basis points compared to low-risk categories. This differential appears justified by risk, but the analysis suggests potential for more granular risk-based pricing that could better align fees with actual fraud exposure.

## Fraud Patterns and Risk Management

The fraud analysis reveals significant variation across merchant categories, with fraud rates ranging from 1.2% to 18.7%. The highest fraud rates appear concentrated in specific merchant categories (5812 and 5942), suggesting that these sectors may benefit from enhanced fraud prevention measures. Temporal patterns in fraud show a 12% increase in fraudulent transactions during holiday periods, suggesting that seasonal fraud prevention measures may be warranted. The data also reveals that credit card transactions have a 23% higher fraud rate than debit card transactions, highlighting the importance of differentiated fraud controls.

The analysis of fraudulent dispute patterns reveals that 62% of fraudulent transactions occur during peak processing hours (10:00-15:00), suggesting that real-time fraud detection systems may benefit from enhanced monitoring during these periods. The data also shows that merchants with integrated fraud prevention systems experience 37% lower fraud rates compared to those relying solely on basic controls, highlighting the value of advanced fraud prevention investments.

## Merchant Performance and Operational Patterns

The merchant performance analysis reveals significant variation in operational patterns across merchant categories. The top-performing merchants by transaction volume process an average of 487 transactions per day, compared to the overall average of 423 transactions. Merchant category 7997 demonstrates particularly strong performance, with 28% higher transaction volumes than the category average. The data shows that merchants with integrated payment processing systems experience 19% higher transaction volumes compared to those using basic payment solutions.

The analysis of merchant category performance reveals that categories 7993 and 7372 demonstrate particularly strong performance metrics, with 22% and 18% higher average transaction values respectively. These categories also show lower fraud rates (4.1% and 5.3%) compared to the overall average of 6.3%, suggesting that certain business models may inherently carry lower fraud risk. The data also reveals that merchants with dedicated payment processing teams experience 27% lower chargeback rates compared to those managing payments in-house.

## Technical Infrastructure and System Performance

The technical analysis of payment processing systems reveals significant variation in system performance across merchant categories. Transactions processed through integrated payment platforms experience 18% faster settlement times compared to those using basic payment solutions. The data shows that merchants with API-integrated payment systems experience 23% fewer failed transactions compared to those using basic payment gateways. The analysis reveals that system performance varies significantly by merchant category, with category 7997 demonstrating particularly strong system performance metrics.

The analysis of system performance during peak periods reveals that merchants with load-balanced processing systems experience 31% fewer system failures during peak hours compared to those using basic infrastructure. The data shows that merchants with real-time payment processing systems experience 19% faster settlement times compared to those using batch processing models. The analysis also reveals that merchants with integrated fraud prevention systems experience 27% fewer chargebacks compared to those relying solely on basic fraud controls.

## Strategic Recommendations and Conclusions

Based on the comprehensive analysis, we recommend that payment processors develop category-specific fee structures that account for the varying risk profiles and operational characteristics of different merchant categories. The data supports the development of risk-based pricing models that incorporate merchant category, transaction patterns, and historical fraud rates. The analysis suggests that implementing category-specific fraud prevention measures could reduce overall fraud costs by an estimated 18-22%. The findings also support investment in advanced fraud prevention technologies, particularly for high-risk merchant categories.

The analysis suggests that payment processors should develop category-specific performance benchmarks to help merchants optimize their payment processing operations. The data supports the development of category-specific best practice guides that address the unique challenges of different merchant categories. The findings also suggest that payment processors should develop category-specific support resources to help merchants optimize their payment processing operations. The comprehensive nature of this analysis provides a foundation for ongoing research into payment processing optimization and fraud prevention strategies. | Here's the report summarizing the notable trends and findings from the data analysis:

### 1. Payments Data Summary (payments.csv)
- **Size**: 23,028.7 KB (largest dataset)
- **Rows**: 10,000 (sampled for analysis)
- **Columns**: Found in dataset (specific names not available)
- **Findings**:
- Contains both numerical and categorical columns
- Some categorical columns have distinct value distributions
- Potential date/time columns identified

### 2. Merchant Category Codes (merchant_category_codes.csv)
- **Size**: 26.0 KB
- **Count**: Specific count not available
- **Findings**:
- Contains merchant category codes (MCC)
- Some merchant codes in payments data not found in category file

### 3. Acquirer Countries (acquirer_countries.csv)
- **Size**: 0.2 KB
- **Count**: Specific count not available
- **Findings**:
- Contains country information for payment acquirers
- Shows distribution of top countries

### 4. Merchant Data (merchant_data.json)
- **Size**: 6.7 KB
- **Count**: Specific count not available
- **Findings**:
- Contains merchant information with identifiable keys
- Some merchant IDs in payments data not found in merchant data

### 5. Fees Structure (fees.json)
- **Size**: 518.7 KB
- **Record Count**: Specific count not available
- **Findings**:
- Contains fee information with identifiable keys
- Some acquirer IDs in fees data not found in countries file

### 6. Documentation
- **Manual**: Present (manual.md, 21.6 KB)
- **Readme**: Present (payments-readme.md, 1.7 KB)

### Key Observations:
1. **Data Relationships**:
- Potential linkage issues between datasets (merchant IDs, category codes, and acquirer IDs)
- Merchant category codes in payments data don't fully match category reference file
- Merchant IDs in payments data don't fully match merchant data records
- Acquirer IDs in fees data don't fully match countries file

2. **Data Structure**:
- Payments data appears to be the core transactional dataset
- Supporting files provide reference information (categories, countries, merchants, fees)
- JSON files contain structured data while CSVs contain tabular data

3. **Data Quality Notes**:
- Inconsistencies between data across different datasets
- Potential data integration challenges due to ID mismatches
- Documentation available to assist with interpretation

The analysis reveals the payments dataset as the central component, with supporting files providing contextual information. Notable data integrity issues were identified regarding ID matching across datasets, which should be addressed before further analysis. | Below is the summary report based solely on the available information (i.e. file names, sizes, and expected content types):

Report on Data Files in Payment Dataset

1. Overview of the Dataset Files:
• acquirer_countries.csv (≈0.2 KB): A small CSV that likely contains metadata or mappings for acquiring country codes or names.
• fees.json (≈518.7 KB): A moderately sized JSON file that probably provides detailed fee structures, fee types, or fee calculations applicable to the payment transactions.
• manual.md (≈21.6 KB): A markdown document serving as a manual. This file is expected to include context, instructions, or detailed explanations regarding how to work with or interpret the dataset.
• merchant_category_codes.csv (≈26.0 KB): A CSV containing codes that classify merchants. This file may help payment transactions (or fee calculations) to business categories.
• merchant_data.json (≈6.7 KB): A relatively small JSON file likely containing supplementary information about merchants (such as names, IDs, or basic attributes).
• payments-readme.md (≈1.7 KB): A short README file that probably documents the structure, source, and nuances of the payments dataset.
• payments.csv (≈23028.7 KB): By far the largest file in the set (over 23 MB). This appears to be the core transactional dataset, probably containing detailed records of payments, such as timestamps, merchant identifiers, amounts, transaction types, etc.

2. Notable Trends & Findings Based on File Characteristics:
• Main Data Concentration: The payments.csv file is significantly larger than all other files. This indicates that while several supporting documents and lookup files exist, the primary volume of data and most likely the majority of trends or analytics will come from analyzing this payment transactions file.
• Structured Supporting Information: The presence of several lookup files (acquirer_countries.csv, merchant_category_codes.csv, and merchant_data.json) suggests that the transactions in payments.csv are cross-referenced against standardized metadata. This design is typical of well-organized transactional databases, where linking transactions to merchant or fee information is necessary.
• Rich Fee Data: The fees.json file, at over half a megabyte, indicates that detailed fee structures (perhaps by transaction type, country, or merchant category) are maintained. Comparisons between transaction records and fee specifications may reveal trends such as fee variations across countries or merchant categories.
• Complementary Documentation: The two markdown files (manual.md and payments-readme.md) indicate that the dataset is documented thoroughly. Such documentation is valuable in understanding the relationships between files, expected formats, and proper analytical approaches.
3. Potential Analytical Approaches (if full file contents were to be explored):
• Analyzing payments.csv to extract transaction patterns (volumes over time, frequency per country/merchant, seasonal spikes, etc.).
• Linking fees from fees.json with transactions to determine profitability, fee patterns, or cost structures.
• Utilizing merchant_category_codes.csv and merchant_data.json to build categorical analyses (e.g., trends in industries, high-activity merchant groups).
• Validating or cross-referencing acquirer country data with transaction records to study geographic distribution or country-specific behaviors.
• Consulting the manual.md and payments-readme.md documents to understand potential data quality issues, processing instructions, or additional attributes not immediately visible from the file names.

Conclusion:
Based on the provided file characteristics, the dataset is centered around a large volume of payment transactions that are supported by detailed fee structures, merchant classifications, and country mappings. The thorough documentation implies that the dataset is intended for in-depth analytical studies, which might include transaction volume trends, fee impact analyses by country or merchant segment, and overall payment behavior. Future analysis, when leveraging the actual contents of these files, can validate these insights and uncover additional detailed findings.

This summary report relies only on file names, sizes, and inferred content types, as no direct content analysis was performed. |

*Figure 15.* A case of autonomous data science for fully open-ended data research.

