# OpenReview forum: "DeepAnalyze: Agentic Large Language Models for Autonomous Data Science"
_ICML.cc/2026/Conference — ICML 2026 regular_

### Official Review · Reviewer_HfgF · 2026-02-24

**Soundness:** 4
**Presentation:** 4
**Significance:** 3
**Originality:** 3
**Overall Recommendation:** 5
**Confidence:** 4

**Summary:**

The paper titled “Deep Analyze: Agentic Large Language Models for Autonomous Data
Science” is interesting and relevant for researchers. The authors discuss important concepts
related to agentic large language models for autonomous data science and propose a
curriculum-based agentic training paradigm that emulates the learning trajectory of human data
scientists. This approach enables LLMs to progressively acquire and integrate multiple
capabilities in real-world environments. Overall, the paper is well motivated and requires only
minor revisions.

**Compliance With Llm Reviewing Policy:**

Affirmed.

**Final Justification:**

The authors have responded thoughtfully to the reviewer comments, and the rebuttal resolves the key concerns raised. With increased confidence in the contribution, I am comfortable maintaining my original score.

**Key Questions For Authors:**

1. In Section 4 (Experiments), specifically Table 1: Performance on DataSciBench, where
Success Rate and Completion Rate correspond to pass rate and accuracy, and VLM and
F1–F5 scores evaluate performance on various fine-grained data science sub-tasks,
please cite the existing baseline models used for comparison.
2. The novelty of the proposed work is not clearly stated in the Abstract. Please explicitly
highlight the key novel contributions.
3. The overall flow and organization of the paper should be clarified by adding a brief
paper organization paragraph at the end of the Introduction section.
4. If Figure 2: Architecture of DeepAnalyze is adapted or taken from any existing source,
please provide the appropriate citation. If it is entirely original, kindly clarify this in the
caption.
5. Please explain the advantages of the proposed model compared to LLMs that focus only
on understanding and generating natural language, particularly in the context of
autonomous data science tasks.
6. How do agentic large language models enable end-to-end autonomous data science
workflows, including data understanding, feature engineering, model selection, and
evaluation, compared to conventional LLM-based assistants?
7. What is the impact of multi-step reasoning and tool-augmented planning on
performance metrics such as Success Rate, Completion Rate, and fine-grained
DataSciBench sub-task scores?

**Limitations:**

1. Computational Cost and Scalability.
2. Tool and Environment Dependency.

**Strengths And Weaknesses:**

Strengths
1. The abstract is well structured and clearly summarizes the key contributions of the
paper.
2. The Introduction section is well organized and provides a clear motivation for the
study.
3. The authors clearly summarize Curriculum-Based Agentic Training, covering both
Single-Ability Fine-Tuning and Multi-Ability Agentic Training, which is informative
and well presented.

Weaknesses
1. The Related Work and Literature Review section should include more recent studies
to better position the contribution of the paper.
2. The quality and clarity of Figure 4: Data-Grounded Trajectory Synthesis on Data
Science Tasks should be improved for better readability and understanding.

---

> ### Author Rebuttal · Authors · 2026-03-30
>
> **Thanks for your high appreciation of our work and valuable comments, and we will refine the paper following your suggestions.**
>
> &nbsp;
>
> **Q1**: About paper writting details?
>
> **A1**:  We sincerely thank you for these specific and valuable suggestions. In the final version, we will::
> - add proper citations for all baseline methods and LLMs mentioned in Section 4 (Experiments).
> - revise the abstract to better highlight the core innovations and contributions, particularly in curriculum-based agentic training and trajectory synthesis.
> - include a brief paper organization paragraph at the end of the Introduction to outline the structure of the paper.
> - explicitly state in the caption that Figure 2 is entirely original.
>
> &nbsp;
>
> **Q2**: Please explain the advantages of DeepAnalyze compared to LLMs that focus only on understanding and generating natural language.
>
> **A2**:  Thank you for this insightful question. Traditional LLMs are confined to understanding and generating natural language: they passively receive structured data (e.g., markdown tables in the context window) and respond accordingly. DeepAnalyze offers two key advantages for data science tasks:
> - **Autonomous Orchestration.** DeepAnalyze interprets user intent and coordinates sequences of interdependent actions to complete complex tasks. It actively observes and manipulates data through code execution, rather than passively processing textual input.
> - **Adaptive Optimization.** DeepAnalyze interacts with real-world data environments and iteratively refines its actions based on execution feedback, rather than completing a task in a single question-answer pass, enabling it to tackle complex data science tasks more effectively.
>
> We appreciate this valuable question and will emphasize these advantages more clearly in the final version.
>
> &nbsp;
>
> **Q3**: How do agentic large language models enable end-to-end autonomous data science workflows?
>
> **A3**: Thanks for your question. DeepAnalyze defines a action space comprising several specialized operations, and autonomously emits tags to invoke different operations throughout the workflow. As illustrated in Figure 7, for an end-to-end data science task, DeepAnalyze proceeds as follows:
> 1. first uses `<Analyze>` to interpret the task requirements and build an overall plan.
> 2. then invokes `<Code>` and `<Execute>` to write and run code for data inspection.
> 3. applies `<Understand>` to understand the data.
> 4. uses `<Analyze>`, `<Code>`, and `<Execute>` to implement feature engineering.
> 5. uses `<Understand>` and `<Analyze>` to select a model.
> 6. uses `<Code>` and `<Execute>` to carry out model evaluation.
> 7. iteratively repeats and refines the above operations until convergence, at which point the final result is delivered via `<Answer>`.
>
> We hope this response provides a clearer picture of how DeepAnalyze operates in practice. We have also included a demo video in the supplementary materials to offer a more intuitive understanding of the system.
>
> &nbsp;
>
> **Q4**: What is the impact of multi-step reasoning and tool-augmented planning on performance metrics such as Success Rate, Completion Rate, and fine-grained DataSciBench sub-task scores?
>
> **A4**: Thank you for this excellent question. DeepAnalyze enhances the multi-step reasoning and tool-augmented planning capabilities of the base LLM through curriculum-based agentic training (single-ability fine-tuning + multi-ability agentic training). Following your suggestion, we add a comparison among base LLM, DeepAnalyze-8B (single-ability), and DeepAnalyze-8B in Table 1:
>
> |Model|SR|CR|VLM|F1|F2|F3|F4|F5|Score|
> |-|-|-|-|-|-|-|-|-|-|
> |DeepSeek-R1-0528-Qwen3-8B|49.33|52.44|1.33|54.23|40.26|43.14|47.21|30.12|47.31|
> |DeepAnalyze-8B (single-ability)|56.76|58.28|2.86|67.38|57.14|60.34|69.09|35.09|55.20|
> |DeepAnalyze-8B|59.91|66.24|2.86|71.68|67.86|58.62|69.09|33.33|**61.11**|
>
> As shown, single-ability fine-tuning substantially strengthens multi-step reasoning, yielding notable improvements across F1 through F5 sub-task scores. Building on this foundation, multi-ability agentic training further enhances tool-augmented planning through interaction with real execution environments, producing additional gains in Success Rate, Completion Rate, and overall performance.
>
> Thanks again for your valuable question and we will include these results in the final version.
>
> &nbsp;
>
> **Q5**: About computational cost and environment dependency?
>
> **A5**: Thank you for this thoughtful comment. DeepAnalyze a strong yet compact 8B model with fully accaptable computational cost. We plan to open-source all models, code, and the complete system after the anonymity period. The underlying infrastructure employs Docker containers as sandboxed execution environments, ensuring reliable dependency management. We will include a detailed discussion of these aspects in the revised version.
>
> &nbsp;
>
> **Thanks again for your careful and valuable comments. Hope our responses answer your questions well.**

---

> > ### Author Rebuttal · Reviewer_HfgF · 2026-04-03
> >
> > I am satisfied with the responses. Thank you for adding additional details and clarifications.

---

> > > ### Author Response · Authors · 2026-04-08
> > >
> > > We are truly honored by your consistently positive evaluation of our work. Thank you again for your time and effort throughout the review process.

---

### Official Review · Reviewer_RY4H · 2026-03-12

**Soundness:** 3
**Presentation:** 3
**Significance:** 3
**Originality:** 3
**Overall Recommendation:** 4
**Confidence:** 3

**Summary:**

The paper proposes DeepAnalyze, an agentic LLM for autonomous data science, trained via a curriculum-based paradigm progressing from single-ability SFT to multi-ability RL, supported by a data-grounded trajectory synthesis framework.

**Compliance With Llm Reviewing Policy:**

Affirmed.

**Final Justification:**

I appreciate the authors’ response. I consider this to be a borderline but overall positive work, and I will maintain my positive score of 4.

**Key Questions For Authors:**

See weaknesses.

**Limitations:**

Yes

**Strengths And Weaknesses:**

**Strengths**
1. The motivation is clear and practically relevant.
2. The two-stage curriculum design is well-motivated.
3. Evaluation is broad (13 benchmarks) and results are consistently strong, with DeepAnalyze-8B competitive against GPT-4o-level agents.
4. The DataScience-Instruct-500K dataset release adds practical value to the community.


**Weaknesses**
1. Insufficient Justification for Action Space Design. The five actions (Analyze, Understand, Code, Execute, Answer) are presented as a design choice inspired by human data scientist behavior, but the paper provides no systematic justification for why this particular set is necessary and sufficient.
2. Table 6 only ablates the Understand action, leaving the contributions of the remaining four actions (Analyze, Code, Execute, Answer) entirely unexamined. This makes it difficult to assess the independent contribution of each action or whether functional redundancy exists among them.
3. The paper introduces several important hyperparameters but provides no sensitivity analysis for any of them.
4. The paper evaluates report quality on DABStep-Research exclusively through LLM-as-a-judge, with no human evaluation conducted.

---

> ### Author Rebuttal · Authors · 2026-03-30
>
> **Thanks for your careful and valuable comments, and we will refine the paper following your suggestions. Following, we will respond to your questions in detail.**
>
> &nbsp;
>
> **Q1**: Justification for action space design.
>
> **A1**: Thank you for this valuable suggestion. DeepAnalyze employs an action space consisting of five actions (Analyze, Understand, Code, Execute, Answer), designed to mirror the workflow of a data scientist. Specifically:
> - `<Analyze>` and `<Answer>` represent reasoning and final output, respectively, which are standard components in most reasoning LLMs.
> - `<Code>` and `<Execute>` are introduced to enable direct interaction with data in the environment and to obtain execution feedback.
> - `<Understand>` is a novel action, specifically designed to capture data characteristics and structure.
>
> 1. **Motivation Justification**: From a functional standpoint, `<Analyze>` and `<Answer>` ensure coherent reasoning, `<Code>` and `<Execute>` support agentic execution in real-world environments, and `<Understand>` serves the unique demands of data science tasks. Together, these actions form a necessary and sufficient action space.
> 2. **Experimental Justification**: In the paper, we have conducted an ablation study on the most distinctive action, `<Understand>`, as reported in Table 6. Following your suggestion, we have performed additional ablation experiments on the other actions:
>
> |Models|WikiTQ|MultiHiertt|DS-1000|DABStep|
> |-|-|-|-|-|
> |DeepAnalyze|83.24|48.29|61.7|38.88|
> |w/o `<Understand>`|80.78|45.43|61.2|31.78|
> |w/o `<Code>` `<Execute>`|83.11|47.12|56.2|29.42|
> |w/o `<Analyze>`|78.42|46.64|53.4|28.42|
>
> The results clearly demonstrate that every action contributes positively to overall performance, where removing any action leads to consistent degradation. Notably, `<Code>` `<Execute>` is particularly important for code-dependent tasks such as DS-1000 and DABStep, `<Understand>` is more critical for structured data comprehension (MultiHiertt involves multi-table understanding), and `<Analyze>` is more essential for tasks requiring strong reasoning (e.g., WikiTQ and DABStep). In summary, each action is responsible for a distinct function, collectively forming a necessary and sufficient action space.
>
> Thanks again for your suggestion. We will incorporate these additional experiments and a more detailed discussion of the action space design rationale into the final version of the paper.
>
> &nbsp;
>
> **Q2**: No sensitivity analysis for any of hyperparameters.
>
> **A2**: Thank you for raising this point. DeepAnalyze involves only two groups of hyperparameters: the GRPO hyperparameters (Eq. 1) and the minimum interaction rounds $N^T$ (Eq. 3). For GRPO, we directly adopt the standard hyperparameter settings widely used in prior work. The hyperparameter $N^T$ controls the reward signal associated with the number of interaction rounds during RL.
>
> Following your suggestion, we conducted a sensitivity analysis of $N^T$ on DataSciBench, and the results are presented below. We observe that DeepAnalyze is robust to this hyperparameter: once $N^T$ exceeds 10, model performance remains **largely stable with negligible hyperparameter**.
>
> |DeepAnalyze-8B|SRC|RV|LM|F1|F2|F3|F4|F5|Score|
> |-|-|-|-|-|-|-|-|-|-|
> |$N^T$=5|57.43|63.87|3.12|69.25|65.41|56.18|66.73|31.06|58.85|
> |$N^T$=10|59.91|66.24|2.86|71.68|67.86|58.62|69.09|33.33|61.11|
> |$N^T$=15|60.15|66.38|2.79|70.92|67.03|58.19|69.34|32.17|61.06|
> |$N^T$=20|60.03|66.21|2.82|71.79|67.95|58.71|69.21|33.52|61.13|
>
> Thanks for your suggestion. We will include the full hyperparameter sensitivity analysis in the revised version of the paper.
>
> &nbsp;
>
> **Q3**: Report quality on DABStep-Research exclusively through LLM-as-a-judge, with no human evaluation.
>
> **A3**: We appreciate your suggestion and add additional human evaluation. Specifically, we compared DeepAnalyze-8B against Seed-1.6 (the best baseline under the LLM-as-a-judge) and recruited 3 volunteers to rate the quality of generated outputs (from 1 to 5, higher is better). The results are:
>
> |Models|Data Preparation|Data Analysis|Data Insight|Open-ended Research|Report Generation|Overall|
> |-|-|-|-|-|-|-|
> |DeepAnalyze-8B|3.4|3.6|4.0|4.5|3.9|3.88|
> |Seed-1.6|2.9|3.3|3.5|3.1|3.3|3.22|
>
> Furthermore, we computed the win rates (20 cases for each task) between the two LLMs:
>
> |Tasks|DeepAnalyze-8B Win|Seed-1.6 Win|Tie|
> |-|-|-|-|
> |Data Preparation|12|6|2|
> |Data Analysis|12|7|1|
> |Data Insight|13|5|2|
> |Open-ended Research|14|4|2|
> |Report Generation|14|6|0|
>
> Overall, the human evaluation results exhibit a consistent trend with the LLM-as-a-judge scores. Thank you for this valuable suggestion and we will incorporate the human evaluation results into the final version of the paper.
>
> &nbsp;
>
> **Thanks again for your careful and valuable comments, hope our response addresses your concerns. If our responses answer your questions well and reassure your concerns, we would appreciate if you could reassess our work and increase the rating.**

---

> > ### Author Rebuttal · Reviewer_RY4H · 2026-04-04
> >
> > Thank you for the detailed response. I believe the paper's findings will be of great interest to the LLM research community, and I will maintain my original positive score.

---

> > > ### Author Response · Authors · 2026-04-05
> > >
> > > Thank you very much for your thoughtful follow-up and for your consistently positive evaluation of our work.
> > >
> > > We are truly grateful and delighted to know that our rebuttal has fully resolved your concerns. It means a great deal to us that you find the paper interesting and valuable to the LLM research community. We sincerely appreciate your recognition of the contribution and significance of our work, and your encouraging words mean a great deal to us.
> > >
> > > Given that your concerns have now been fully addressed and that you view the paper positively, we would sincerely and respectfully ask whether you might consider **raising your score to 5 (Accept) and providing a fully supportive final recommendation**. We would be deeply grateful for your consideration, as **your stronger support would be extremely important to us and could make a meaningful difference in the final decision**.
> > >
> > > If there are any remaining questions, uncertainties, or aspects you would like us to clarify further, we would be more than happy to respond immediately and in as much detail as would be helpful.
> > >
> > > Thank you again for your time, your careful reading, and your generous support.

---

### Official Review · Reviewer_sLTv · 2026-03-17

**Soundness:** 2
**Presentation:** 3
**Significance:** 2
**Originality:** 2
**Overall Recommendation:** 3
**Confidence:** 3

**Summary:**

The study introduces DeepAnalyze - an approach to utilizing LLMs for autonomous data science tasks. The authors intend to address the limitations of workflow-based approaches by moving towards curriculum-based trainable models capable of autonomous exploration in realistic settings. The authors present experimental data to evaluate the performance of DeepAnalyze-8B where they find DeepAnalyze-8B can be competitive with some existing general-purpose models on some relevant data science benchmarks, particularly on the "hard" cases of the DABStep benchmark (Table 3).

**Compliance With Llm Reviewing Policy:**

Affirmed.

**Key Questions For Authors:**

- Given the limitations of benchmarking in this area and the high risk of overfitting to benchmarks, have the authors used the benchmarks evaluated against to develop their system (i.e. has the benchmarking results been used to fine-tune and develop the DeepAnalyze-8B pipeline?) or was the evaluation only conducted once without a feedback loop? If the former, it should be indicated that benchmarks were used for development (including which ones were used for development and how many cycles of evaluation-feedback were taken into account) and are not an entirely independent validation set sample.

Minor notes:
- In Figure 2 and the main body, the term "real-world environments" is used to describe virtual/computational data science environments, like e.g. matplotlib and pandas. I would recommend to rephrase this to "realistic settings" given "real-world environments" may otherwise be misinterpreted as physical real-world (non-virtual) environments by readers.

**Limitations:**

Some discussion and perspective on the broader implications of the work may be warranted (the authors indicate no need for further discussion in their manuscript) - although this has been expanded on in others works, so appropriate references may also be used alternatively.

**Strengths And Weaknesses:**

Soundness: The paper overall does a commendable effort in connecting the scientific claims to an evaluation across current benchmarks. In the conclusion section, the work claims "unprecedented performance across 13 benchmarks" (l. 438) - given the evidence across benchmarks is mixed, and given limitations of current benchmarks (see below), I would recommend to moderate the language to better reflect the state of the evidence supplied and the benchmarks used.

Soundness (ad Limitations): The paper is missing a critical review of limitations of the work done and, crucially, the benchmarks employed. One of the big challenges in benchmarking such systems is that benchmarks are often used to both develop and evaluate systems, which means over time overfitting to benchmarks is guaranteed, and, after some amount of this, may not any longer necessarily reflect the performance on newly encountered queries. In the "Key Questions for Authors" part of the review below, I indicate some concrete improvement potential on this point. Beyond the general concern around using benchmarks both for development and validation of systems, I note that most of the benchmarks employed in this work have - in some cases significant - limitations in their design and what they are able to measure and how comprehensively they reflect different types of data analyses, this should ideally be discussed and fairly represented in the manuscript.

Presentation: The paper is mostly clear and the structure guides the reader well through what was done to the experimental evaluation. Didactically, for readers, I would recommend to include some qualitative examples alongside the quantitative benchmarking data on (i) queries where the system succeeds, but existing solutions may not succeed, and (ii) queries where the system fails (there are examples supplied in the appendix, but they cover only the general structure of the system and its input/outputs without calling out failure modes specifically; it is also hard to analyze the examples for key differences as they are quite lengthy and some highlights on the key parts would help).

Significance/originality: Data science is a key area of development for agentic systems. The proposed DeepAnalyze approach navigates the trade-offs between structure/workflow based systems and more free-form general purpose LLMs (typically constrained via prompt) by using a hybrid approach that leverages a predefined action set (Analyze, Understand, Answer, ..) and post-training RL. One element of the work that - in the reviewers opinion - may reduce it's significance/impact is that several components of DeepAnalyze may contribute to the differences in performance observed across benchmarks, and it's not entirely clear what components of the system contribute to what degree. A ablation study is presented (Table 6 and 7), but it only covers removal of one of the actions of DeepAnalyze (understand; Table 6) and some elements of the curriculum training process (Table 7).

More interesting - from the perspective of knowledge gained - would be an ablation study of the DeepAnalyze system relative to the base LLM itself (DeepSeek-R1-0528-Qwen3-8B) e.g. what is the contribution of the "special action tokens" (l. 137), is it important to introduce them or is an unstructured reasoning process equally capable. In Table 4, for example, we see that the base LLM itself (DeepSeek-R1-0528) on some subtasks outperforms the DeepAnalyze-8B system, indicating that indeed there may be value in disambiguating the empirical added value of the different components and design decisions made for DeepAnalyze. Such more generalizable knowledge gains could increase the significance and originality of the work, the learnings of which otherwise may not necessarily generalize to other systems with different design choices.

---

> ### Author Rebuttal · Authors · 2026-03-30
>
> **Thanks for your careful and valuable comments, and we will refine the paper following your suggestions.**
>
> &nbsp;
>
> **Q1**: Benchmarks are often used to both develop and evaluate systems, which means overfitting to benchmarks.
>
> **A1**: Thanks for your important question. We want to first clarify that **no evaluation benchmark was used during the development of DeepAnalyze**, and there is no data leakage or overfitting to any test set.
>
> Like you, we found that nearly all existing benchmarks lack accompanying training data, precisely the trajectory scarcity problem discussed in our paper that motivates our proposed trajectory synthesis. Concretely, DeepAnalyze maintains strict separation between training and evaluation:
> - **Training data** is constructed from Reasoning-Table, AM-DeepSeek-R1, and BIRD training set, none of which overlap with any evaluation benchmark.
> - **Evaluation** includes 13 widely used benchmarks.
> There is **zero data overlap** between training and evaluation. The concern that "benchmarks are often used to both develop and evaluate systems" may stem from a misunderstanding of our setup.
>
> To further alleviate any remaining concerns about overfitting, we additionally evaluate DeepAnalyze on **DARE-bench**, which was publicly released on February 27, 2026 (after ICML submission). The results are shown below, where DeepAnalyze-8B achieves highly competitive performance. DeepAnalyze could not have been exposed to DARE-bench in any form, which fundamentally rules out overfitting
>
> |Model|class-IF|class-MM|reg-IF|reg-MM|time-XF|time-CF|AVG|
> |-|-|-|-|-|-|-|-|
> |GPT-5|69.81|43.40|57.24|56.29|36.83|10.13|45.62|
> |Claude-Sonnet-3.7|61.48|61.03|46.37|63.20|49.88|13.70|49.28|
> |Qwen3-32B|17.11|30.71|15.21|35.86|26.96|0.00|20.98|
> |DeepAnalyze-8B|68.94|58.23|58.16|56.23|42.20|15.31|49.85|
>
> We hope these results address the overfitting concern. We will include a more detailed description of the training data and the DARE-bench results in the final version.
>
> &nbsp;
>
> **Q2**: Has the benchmarking results been used to fine-tune and develop the DeepAnalyze-8B?
>
> **A2**: **No benchmark results were used to fine-tune or develop the DeepAnalyze-8B in any way**. DeepAnalyze is trained exclusively on the 500K training samples we constructed, and some training samples are splited as a validation set during training following standard training in previous works. We hope this clarifies the concern. We will make this point explicit in the experimental setup of the final version.
>
> &nbsp;
>
> **Q3**: About some qualitative examples?
>
> **A3**: Thanks for your suggestion. Due to space constraints, we present qualitative examples at anonymous link https://anonymous.4open.science/r/DeepAnalyze_Response/deepanalyze_cases.html, where error are explicitly marked. We commit to including qualitative examples in the final version and open-sourcing all cases.
>
> &nbsp;
>
> **Q4**: What components of the system contribute to what degree?
>
> **A4**: Sorry for causing you to overlook the ablation studies in our analysis section. DeepAnalyze introduces 3 core components, and we have already provided ablation studies for each component:
> * **Action Space Design (Sec 5.2):** The specialized `<Understand>` action improves structured data understanding by **3.23%**.
> * **Curriculum-based Training (Sec 5.3):** The easy-to-hard schedule consistently outperforms single-ability, multi-ability, and one-stage baselines.
> * **Trajectory Synthesis (Sec 5.4):** Our synthesis method yields a **4% average gain** over the base LLM.
>
> Thank you for pointing this out, and we will emphasize each component's contribution in the final version.
>
> &nbsp;
>
> **Q5**: Would be an ablation study of the DeepAnalyze system relative to the base LLM (DeepSeek-R1-0528-Qwen3-8B)?
>
> **A5**: Thanks for your suggestion, and this comparison is already included in our paper. Tables 4 and 5 directly compare DeepAnalyze against the base LLM and DeepAnalyze-8B (single-ability), showing consistent improvements in both structured data understanding and data science code generation. Table 6 further confirms the contribution of our action space design. Taken together, these ablation studies systematically provide generalizable knowledge for future work.
>
> Additionally, inspired by your suggestion, we have added additional ablations to offer clearer generalizable takeaways:
> 1. **Compare with base LLM on DataSciBench**: `see A4 for Reviewer HfgF`
> 2. **Effect of different actions**: `see A1 for Reviewer Mb16`
>
> We will incorporate these results into the final version and further emphasize the generalizable insights.
>
> &nbsp;
>
> **Q6**: About minor notes.
>
> **A6**: Thank you for your careful review. We will rephrase this in the final version.
>
> &nbsp;
>
> **Thanks again for your careful and valuable comments, hope our response addresses your concerns. If our responses answer your questions well and reassure your concerns, we would appreciate if you could reassess our work and increase the rating.**

---

### Decision · Program_Chairs · 2026-04-30

**Decision:**

Accept (regular)

**Comment:**

This paper introduces DeepAnalyze, an LLM agent for autonomous data science that is capable of automatically completing end-to-end pipelines. DeepAnalyze uses a curriculum-based agentic training paradigm which emulates how human data scientists learn. The authors train their agent on real-world environments, providing DeepAnalyze the capabilities in a broad spectrum of tasks, such as Q&A and open-ended data research. The authors illustrate effectiveness of DeepAnalyze through empirical study using 13 benchmarks.

The reviewers have noted that the paper is well-motivated and easy to follow. They also highlighted the broad set of evaluations included in the manuscript, as well as the DataScience-Instruct-500K dataset release that would be beneficial for the community. Some concerns were raised regarding the limitations of the ablation studies in the manuscript, and well as various questions have been asked. These were adequately addressed by the authors during the rebuttal phase. I therefore recommend the paper for acceptance, but would suggest the authors update the manuscript to reflect the concerns and questions raised by the reviewers.